# Emergent Neural Network Mechanisms for Generalization to Objects in Novel Orientations

**Avi Cooper**                                                          *acooper@fujitsu.com*
*Fujitsu Research of America*

**Daniel Harari**
*The Institute for Artificial Intelligence*
*Weizmann Institute of Science*
*Center for Brains, Minds and Machines*
*Massachusetts Institute of Technology*

**Tomotake Sasaki**
*Fujitsu Limited*

**Spandan Madan**
*Department of Computer Science*
*Harvard University*

**Hanspeter Pfister**
*Department of Computer Science*
*Harvard University*

**Pawan Sinha**
*Department of Brain and Cognitive Sciences*
*Massachusetts Institute of Technology*

**Xavier Boix**
*Fujitsu Research of America*

**Reviewed on OpenReview:** *https://openreview.net/forum?id=4wBQTZVSHU*

## Abstract

The capability of Deep Neural Networks (DNNs) to recognize objects in orientations outside the training data distribution is not well understood. We investigate the limitations of DNNs' generalization capacities by systematically inspecting DNNs' patterns of success and failure across out-of-distribution (OoD) orientations. We present evidence that DNNs (across architecture types, including convolutional neural networks and transformers) are capable of generalizing to objects in novel orientations, and we describe their generalization behaviors. Specifically, generalization strengthens when training the DNN with an increasing number of familiar objects, but only in orientations that involve 2D rotations of familiar orientations. We also hypothesize how this generalization behavior emerges from internal neural mechanisms — that neurons tuned to common features between familiar and unfamiliar objects enable out of distribution generalization — and present supporting data for this theory. The reproducibility of our findings across model architectures, as well as analogous prior studies on the brain, suggests that these orientation generalization behaviors, as well as the neural mechanisms that drive them, may be a feature of neural networks in general.

# 1   Introduction

Recognizing objects in novel orientations lies at the heart of biological and artificial intelligence, as it is a fundamental capacity necessary to understand the visual world (Sinha & Poggio, 1996; Ullman, 1996). However, the computational mechanisms underlying this capacity in both brains and machines are not yet well understood (Gazzaniga et al., 2006; Pinto et al., 2008; Li et al., 2021).

In the realm of artificial systems, Deep Neural Networks (DNNs) have recently made large strides in object recognition (He et al., 2017; Carion et al., 2020). However recent studies have shown that DNNs perform poorly when objects are presented in novel orientations, even when learning from large datasets with millions of examples (Barbu et al., 2019; Alcorn et al., 2019; Madan et al., 2022; Abbas & Deny, 2023; Ollikka et al., 2025). More broadly, novel orientations are a special case of *out-of-distribution* (OoD) data. DNNs' generalization is often limited to the images from the training distribution, known as *in-distribution* data, while it remains difficult to give a principled account of their performance in OoD settings.

One promising approach to understand the capabilities of DNNs is to leverage the knowledge gained from studying biological intelligence (Hassabis et al., 2017; Ullman, 2019). In natural settings, biological intelligent agents observe instances of object categories from diverse orientations. When encountering a new object instance, these agents often demonstrate the capacity to accurately identify the object in different orientations by drawing upon past experiences with similar instances (Booth & Rolls, 1998; Freiwald & Tsao, 2010; Ratan Murty & Arun, 2015). Extensive investigations into human and mammalian perception and object recognition in unfamiliar orientations have revealed that recognition accuracy varies across novel orientations, with some orientations exhibiting superior generalization compared to others (Logothetis & Pauls, 1995). Additionally, studies into the neural mechanisms underlying these cognitive abilities have marshaled compelling evidence suggesting that neurons respond to their own specific set of object features when present in the visual field (Desimone et al., 1984; Kobatake & Tanaka, 1994; Gauthier et al., 2002; Fang & He, 2005). This neural tuning has been reported to be invariant to a certain degree from the object's orientation (Logothetis & Sheinberg, 1996). Theoretical frameworks have proposed that such neural invariance to object orientation forms the basis for the ability to recognize objects in novel orientations within biological systems (Poggio & Anselmi, 2016).

In this paper we employ these same analytical tools utilized in the study of biological brains in order to understand DNNs' generalization abilities in OoD orientations. Specifically, we study DNNs under conditions akin to the operating regime of biological brains, in which some instances of an object category (*e.g.,* a 'Boeing 777 airliner' is an instance of the 'airplane' category) are seen from all orientations during training (*fully-seen* instances), while other instances are only seen in a subset of all orientations (*partially-seen* instances). During test time, we evaluate the generalization performance of the networks by measuring instance classification performance on OoD orientations (*i.e.,* those orientations not included in the training set) of *partially-seen* instances. This simple paradigm, inspired by (Jang et al., 2023), facilitates analyzing the impact of several key factors that may influence OoD generalization, such as the number of *fully-seen* instances and the *in-distribution* orientations of the *partially-seen* instances. This paradigm allows us to more precisely characterize performance challenges of DNNs for OoD orientations. Figure 1 summarizes the paradigm that we follow in this work.

A large number of previous works have begun to understand the generalization capacities in DNNs to OoD orientations. For example, (Lenc & Vedaldi, 2015; Gruver et al., 2023) investigated the emergence of invariance, and specifically rotational invariance to affine rotations. However, it remains unclear whether and how DNNs generalize to OoD orientations, such as in the task we outlined above. Our novel analytical approaches yield three important findings: 1) DNNs readily generalize to certain surprising OoD orientations. 2) These generalizable orientations are parameterized by the *in-distribution* set, and can be quantifiably predicted. 3) Neural representations at the individual-unit level, across orientations and object instances, support a theory on brain-like mechanisms that drive the emergence of orientation invariance (Logothetis & Sheinberg, 1996; Poggio & Anselmi, 2016).

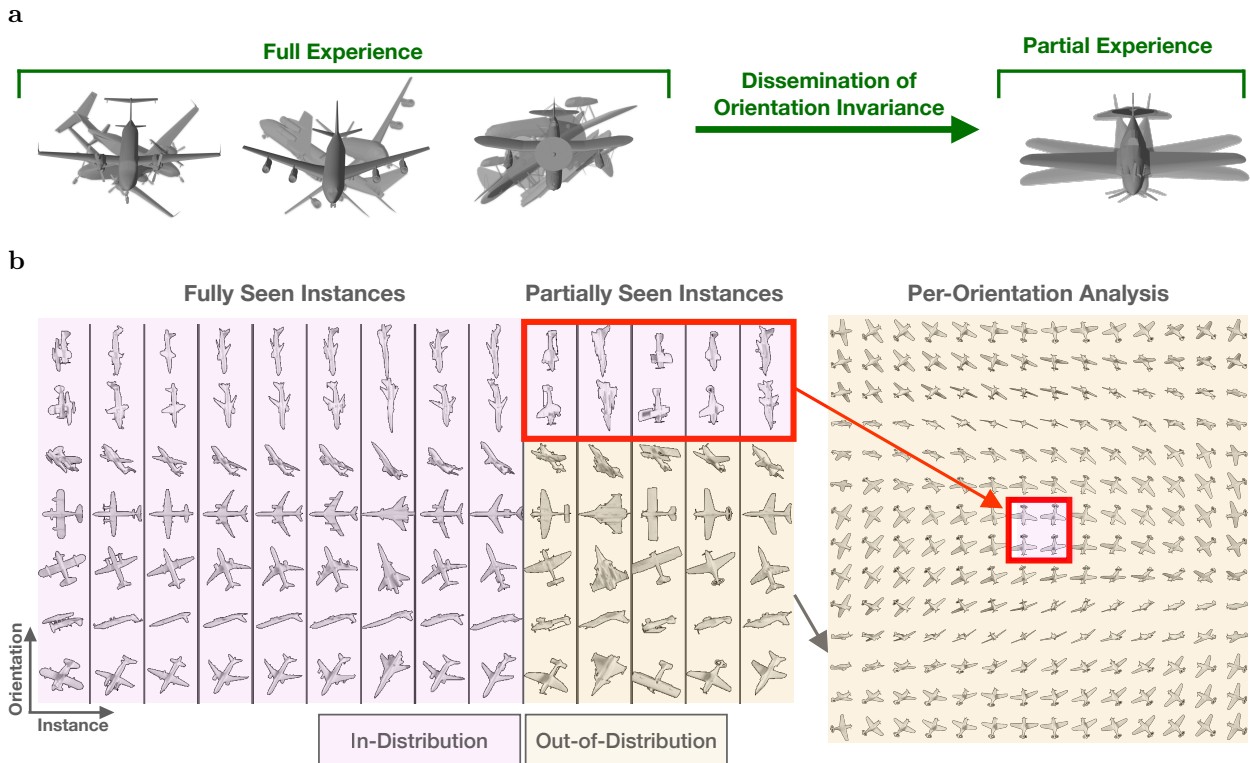

Figure 1: **Learning paradigm and network's per-orientation accuracy. (a)** The network is trained with images of certain airplanes at all orientations, constituting a 'full experience,' and for other airplanes, a small subset of orientations, constituting a 'partial experience.' The *out-of-distribution*, or OoD, generalization capacities of the network are evaluated by measuring the classification accuracy for *partially-seen* airplanes at unseen orientations. Our results suggest that OoD generalization is facilitated by the dissemination of orientation invariance developed for all orientations for the *fully-seen* airplanes to the OoD orientations of the *partially-seen* airplanes. **(b)** Left: The learning paradigm employed in this work. Each column is a sample object instance (here from the airplane dataset) and each row is a sample orientation. The training set includes all orientations for *fully-seen* instances, and a partial set of orientations (outlined in red) for *partially-seen* instances (in this example, with the airplanes' nose pointing down). The orientations included in the training set are referred to as *in-distribution* orientations (pink shading). Orientations of the *partially-seen* instances that are not included in the training set are referred to as OoD (yellow shading). Right: A visualization of per-orientation-analysis. The set of all orientations are arranged to capture proximity relationships between orientations. (Further details are provided in Fig.3a.)

## 2  Methods

In this section we detail the experimental setup and analytical tools we employ in this work, which were first introduced in Section 1 and depicted in Figure 1. In the following, we outline the dataset and DNN architecture details in the next two subsections. We then present the analytical tools employed in Section 3. Specifically, we introduce a visualization of per-orientation accuracy, a predictive model of DNN generalization, and finally a neural metric related to generalization.

Code used for this study can be found at: `https://github.com/avicooper1/OOD_Orientation_Generalization`. Data generated and analyzed in this study can be found at: `https://doi.org/10.7910/DVN/M9WPNR`.

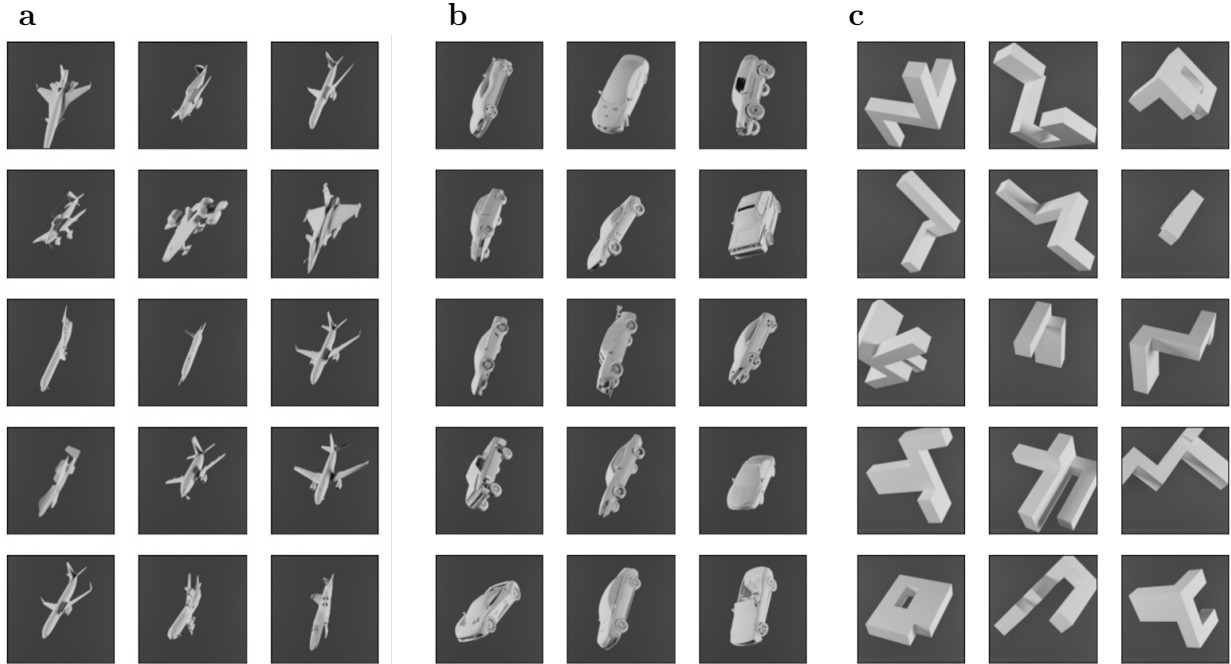

Figure 2: **Object Datasets.** In our experiments we used three object categories: (**a**) Airplanes, (**b**) Cars, and (**c**) Shepard&Metzler objects. The first two were curated from ShapeNet (Chang et al., 2015) and we procedurally generated the last one.

## 2.1 Datasets

The datasets consist of images of object instances, some of which are seen from all orientations during training (*fully-seen*) and other instances which are only seen from a subset of orientations during training (*partially-seen*). Each dataset also contains a validation set of *fully-seen* images and *partially-seen* instances at orientations seen during training. The distributions of the validation and train overlap, but no specific images overlap. The test set consists of images of the *partially-seen* instances at OoD orientations, which are not included during training. For each experiment, we construct a dataset with an object category, a set of seen orientations for the *partially-seen* instances, and a proportion of *fully-seen* to *partially-seen* instances. We describe these elements below.

**Object Categories.** We used three categories of objects: Airplanes, Cars and *Shepard&Metzler* objects. For the airplanes and cars we curated 50 high quality object instances of each category from the ShapeNet(Chang et al., 2015) database. Both airplanes and cars have clear axes of symmetry, which allow for intuition of how networks generalize to OoD orientations. We therefore also experimented with highly asymmetric objects similar to those tested for 3D mental rotations in (Shepard & Metzler, 1971) (which we denote as *Shepard&Metzler* objects; Fig. 2). We procedurally generated the *Shepard&Metzler* objects. Images were rendered from the 3D models under fixed lighting conditions, and the models were centered and fully contained within the image frame.

We experimented with datasets where both *fully-seen* and *partially-seen* instances are from the same category, and also where they come from different categories. For example, where the *fully-seen* instances were airplanes, and the *partially-seen* instances were *Shepard&Metzler* objects. In this case, high classification accuracy on the test set (*i.e.,* the *partially-seen* instances in OoD orientations) requires that orientation generalization disseminates to new object categories.

***In-Distribution* Orientations for *Partially-Seen* Instances.** We chose several ranges of orientation to be *in-distribution* for the *partially-seen* instances, which we refer to as *seed* orientations. We use the term

| $\alpha$ Range | $\beta$ Range | $\gamma$ Range | Seed Name |
|:---:|:---:|:---:|:---:|
| $(-\pi, \pi)$ | $(-0.1, 0.1)$ | $(-0.25, 0.25)$ | $\hat{\alpha}$ |
| $(-0.25, 0.25)$ | $(-\frac{\pi}{2}, \frac{\pi}{2})$ | $(-0.25, 0.25)$ | $\hat{\beta}$ |
| $(-0.25, 0.25)$ | $(-0.1, 0.1)$ | $(-\pi, \pi)$ | $\hat{\gamma}$ |
| $(-\pi, \pi)$ | $(-0.1, 0.1)$ | $(-1.8, -1.3)$ | $\hat{\alpha}'$ |

Table 1: In-distribution orientations, in radians, for *partially-seen* instances, termed *seed* orientations. The *seed* name is chosen for the full axis, *i.e.,* $\hat{\alpha}$ is for the *seed* where $\alpha$ ranges from $-\pi$ to $\pi$.

*seed* because any generalization to OoD orientations must stem from these orientations. In choosing the *seed* orientations, we chose small ranges of rotation along two of the three axes, and the full range of rotation along the final axis. This allowed for some variation in the instance appearance along the "full" axis, while still preserving many OoD orientations along the "bounded" axes. We employed several *seeds* to ablate any idiosyncrasies within any particular choice.

To express an orientation we use $\boldsymbol{\theta} := (\alpha, \beta, \gamma)$, the Euler angles with respect to the orthogonal axes of a reference coordinate system $\mathbb{R}^3$ (Goldstein et al., 2002), with the convention that $\alpha$ and $\gamma$ are bounded within $2\pi$ radians and $\beta$ is bounded within $\pi$ radians. Following this convention, the seeds we employed are enumerated in Table 1. Note that we experiment with four *seeds*, which are denoted as $\hat{\alpha}$, $\hat{\beta}$, $\hat{\gamma}$ and $\hat{\alpha}'$ as they refer to the axis that is fully observed.

**Proportion of *fully-seen* instances.** In each experiment, there were always 50 instances that were seen. However we vary the number of *fully-seen* instances $N$ between 10 (20% of the total number of instances) and 40 (80%). The remaining instances are *partially-seen*. For a fair evaluation of the effect of data diversity, the amount of training examples is kept constant as we vary the data diversity.

**Dataset Size.** Each dataset is 200k images, 4k image for each of the 50 object instances. A training epoch iterates through every image in the dataset once. Since the generalization abilities of the network may change with the amount of data, we evaluate OoD accuracy under the most favorable scenario for the DNN, in which adding more data does not lead to a higher OoD accuracy. We repeat the experiments with half the data, see Figures S5, S6, S8, and results are consistent with our findings for the default dataset size, which provide some reassurance that adding more data won't change the conclusions of this paper.

## 2.2 Experimental Setup

For each experiment, we train a DNN from scratch on a supervised instance classification task (*i.e.,* N-way classification where each dimension of the output layer corresponds to one object instance), using the datasets described above in Sec. 2.1. In this subsection we outline the DNNs experimented with, along with other training parameters.

**DNN Architectures.** We tested ResNet18 (He et al., 2016) on all permutations of dataset structure (combination of *seed*, object category, and proportion of *fully-seen*) as well as for all experimental ablations (pretrained on ImageNet, image augmentations, and using half the training data: see Figs. S4„S5S6,S8). To ensure that our results were not specific to ResNet18, we ran the same experiments with DenseNet121 (Huang et al., 2017), ViT-Base (Vaswani et al., 2017; Dosovitskiy et al., 2020) and CORnet-S (Kubilius et al., 2018). The first two were chosen as they are representative feed-forward DNNs. The architecture of CORnet is brain-inspired and includes recurrence at higher layers in addition to convolutions in lower layers. We observed the same behavior for all analyses for all the architectures when analyzing the results of airplane datasets (across number of *fully-seen*, *seed* orientation etc.) Since we observe results are consistent for all architectures, we only report the full battery with ResNet. For DenseNet and CORnet, we report results with the airplane dataset with all *seeds*. For ViT-Base, we report results with the airplane dataset on the $\hat{\alpha}$ *seed*.

| Architecture | Learning Rate | Batch Size |
|---|---|---|
| 1. ResNet18 | $10^{-3}$ | 230 |
| 2. DenseNet121 | $10^{-3}$ | 64 |
| 3. CORnet-S | $10^{-4}$ | 128 |
| 4. ViT-Base | $10^{-4}$ | 256 |

Table 2: Training Hyperparamters

Figures in the main text are shown only for ResNet, except for Figs. 5b,c, which are scatter plots and therefore don't require averaging over architectures. The analysis was repeated for all architectures and these results can be found in the supplement. This includes accuracy heatmap visualizations (Fig. S3), average OoD accuracy (Fig. S8), and predictive modeling (Fig. S5). The same conclusions drawn in the main paper may be drawn from the results with other architectures as well.

**Hyperparameters for training.** We trained the four deep convolutional neural network architectures using the Adam Optimizer (Kingma & Ba, 2017) with the learning rates and batch sizes listed in Table 2.

Batch sizes were chosen to be as large as possible while still fitting the model, the batch of images and forward-pass computations in memory. Learning rates were chosen from $10^x, x \in \{-1, -2, -3, -4, -5\}$ to be as large as possible while ensuring that validation accuracy, which included *fully-seen* instances from all orientations and *partially-seen* instances from *in-distribution* orientations, remained stable. Each network was trained for a minimum of 100 epochs, after which training was stopped if validation performance was lower than some epoch for seven epoch in a row. After this point *in-distribution* performance was stabilized at 100% and OoD performance reached an asymptote.

**Hardware details.** Experiments were run with one CPU, 25GB of memory and on several generations of Nvidia GPUs with a minimum of 11GB of memory.

**Repetition of the experiment.** We re-run each experiment (tuple of dataset and model architecture) five times, each time randomly sampling the specific instances which comprise the *fully-seen* and *partially-seen* sets. For all figures, data come from all available repetitions, except where otherwise noted. Error bars denote one standard deviation from the mean.

## 2.3 Per-Orientation Accuracy Visualization

Previous works have typically reported average performance over all orientations. In contrast, we evaluate the network's performance for each orientation across the entire range of orientations. To express an orientation of an object instance we use $\boldsymbol{\theta} := (\alpha, \beta, \gamma)$, the Euler angles with respect to the orthogonal axes of a reference coordinate system $\mathbb{R}^3$ (Goldstein et al., 2002), with the convention that $\alpha$ and $\gamma$ are bounded within $2\pi$ radians, and $\beta$ is bounded within $\pi$ radians. We define $\Psi(\boldsymbol{\theta}) \in [0, 1]$ to be the network's average classification accuracy at an orientation $\boldsymbol{\theta} = (\alpha, \beta, \gamma)$ over either the *fully-seen* or *partially-seen* instances.

To facilitate intuition of $\Psi$ we introduce a visual representation of this function. Since orientations are continuous values and are related spatially we map the range of bounded values of orientations $(\alpha, \beta, \gamma)$ onto a Cartesian coordinate system, resulting in a cube—the basis of our visualization. We discretize the continuous space of orientations into *cubelets*, which are sub-cubes with a width of $\frac{1}{\#Cubelets}$ of the full range of each respective angle. This approach preserves local behavior in aggregate analysis. In addition, we outline the range of orientations which are *in-distribution* for the *partially-seen* instances — the rest are OoD orientations. To illustrate the object orientation at a given *cubelet*, we sample one representative image and overlay it onto the heatmap at the location of the *cubelet*.

See Fig. 3a, which shows this visual representation scheme, and Figs. 3b and 3c for examples.

### 2.4 Model of DNN Per-Orientation Generalization

We hypothesize a set of rules which govern the partitioning of the orientation space into *generalizable* and *non-generalizable* orientations. To quantitatively evaluate this hypothesis we formulate a model of the partitioning rules, which can be used to predict the OoD generalization patterns of the network, given a *seed* of *in-distribution* orientations. Briefly, the model, denoted by $f_{\mathbf{w}}(\boldsymbol{\theta})$, has three components: $A(\boldsymbol{\theta})$, which captures small angle rotations around $\boldsymbol{\theta}$; $E(\boldsymbol{\theta})$, which captures in-plane (2D) rotations; $S(\boldsymbol{\theta})$, which captures object "silhouette" projections at the orientation $\boldsymbol{\theta}$. We define $f_{\mathbf{w}}(\boldsymbol{\theta})$ as the predictive model for generalization per each orientation. To measure the goodness of our prediction, we employ the Pearson correlation coefficient to measure how closely our model correlates with DNN recognition accuracy, $\Psi(\boldsymbol{\theta})$. We choose this metric because it normalizes data with respect to amplitude and variance, and therefore measures patterns of behavior across $\boldsymbol{\theta}$ and relative to other $\boldsymbol{\theta}$, rather than the exact performance for every $\boldsymbol{\theta}$.

The model's three components ($A(\boldsymbol{\theta})$, $E(\boldsymbol{\theta})$ and $S(\boldsymbol{\theta})$) easily lend themselves to formalization with Euler's rotation theorem (Goldstein et al., 2002). The theorem states that any rotation can be uniquely described by a single axis, represented by a unit vector $\hat{\mathbf{e}} \in \mathbb{R}^3$, and an angle of rotation, denoted as $\phi \in [0, \pi]$ around the axis $\hat{\mathbf{e}}$. We employ this representation to describe the rotation between an arbitrary orientation of interest, $\boldsymbol{\theta}$, and an orientation in the set of *in-distribution*, denoted $\boldsymbol{\theta}_s \in \Omega_s$. We use $\hat{\mathbf{e}}_{\boldsymbol{\theta},\boldsymbol{\theta}_s}$ and $\phi_{\boldsymbol{\theta},\boldsymbol{\theta}_s}$ to denote the unit vector (axis) and the angle of this rotation, respectively.

**Component 1: Small Angle Rotation, $A(\boldsymbol{\theta})$.** The first component of the model captures orientations that are small angle rotations from the orientations in the training distribution. Visually similar orientations are those that are arrived at by small rotations from *in-distribution* orientations, or small $\phi_{\boldsymbol{\theta},\boldsymbol{\theta}_s}$. We therefore define the first component $A(\boldsymbol{\theta})$ as

$$A(\boldsymbol{\theta}) := \max_{\boldsymbol{\theta}_s \in \Omega_s} \left| 1 - \frac{\phi_{\boldsymbol{\theta},\boldsymbol{\theta}_s}}{\pi} \right| \in [0, 1]. \tag{1}$$

The $\max_{\boldsymbol{\theta}_s \in \Omega_s}$ operator chooses the *in-distribution* orientation that is closest to $\boldsymbol{\theta}$ of interest.

**Component 2: In-plane Rotation, $E(\boldsymbol{\theta})$.** The second component of the model captures orientations which appear as *in-plane* rotations of *in-distribution* images. Let $\mathbf{c} \in \mathbb{R}^3$ be the unit vector representing the camera axis. *In-plane* rotations are those for which the axis of rotation is parallel to the camera axis. Thus, an orientation appear as an *in-plane* rotations of an *in-distribution* images when $\mathbf{c} \in \mathbb{R}^3$ and $\hat{\mathbf{e}}_{\boldsymbol{\theta},\boldsymbol{\theta}_s} \in \mathbb{R}^3$ (*i.e.,* the vector of object instance rotation) are parallel. Taking their standard inner product yields the proximity to being parallel, which is therefore the degree to which the rotation is *in-plane*.

Thus, we define the second component $E(\boldsymbol{\theta})$ as follows:

$$E(\boldsymbol{\theta}) := \max_{\boldsymbol{\theta}_s \in \Omega_s} \left| \mathbf{c}^\top \hat{\mathbf{e}}_{\boldsymbol{\theta},\boldsymbol{\theta}_s} \right| \in [0, 1], \tag{2}$$

where $\mathbf{c}^\top$ denotes the transpose of $\mathbf{c}$.

**Component 3: Silhouette, $S_A(\boldsymbol{\theta})$, $S_E(\boldsymbol{\theta})$.** We refer to the object's silhouette as a featureless solid shape with its edges matching the outline of the object seen as a shadow from the camera view. The third component of the model captures orientations in which the object's silhouette is similar to the silhouette of the object at the *in-distribution* views — for example, the similarity in the appearance of an airplane's silhouette when viewed from above and from below. These orientations are defined as a $\pi$ radians rotation around the $\gamma$ axis, which results in a silhouette orientation. We transform all the *in-distribution* orientations, $\Omega_s$, in this way, and we call these silhouette *in-distribution* orientations $\Omega_{\hat{s}}$. We then compute $S_A(\boldsymbol{\theta})$ and $S_E(\boldsymbol{\theta})$, substituting $\Omega_{\hat{s}}$ for $\Omega_s$ in $A(\boldsymbol{\theta})$ and $E(\boldsymbol{\theta})$ respectively.

**Nonlinearities.** The components described above capture a general trend, but do not match the range of values given by a 0-100% accuracy metric. We therefore fit the components with a logistic function. The

'S'-like shape of the logistic function allows for the highest and lowest values of $E(\boldsymbol{\theta})$, $A(\boldsymbol{\theta})$, $S_A(\boldsymbol{\theta})$ and $S_E(\boldsymbol{\theta})$ to be close to the highest and lowest values of $\Psi(\boldsymbol{\theta})$. In addition, it allows for a smooth transition between these highest and lowest values. Most importantly, the simplicity of the logistic function allows for fitting while preserving the interpretability of the model components, ensuring that the models remains related to small angle, *in-plane* and silhouette rotations. We employ the following logistic function:

$$\sigma(x; (a,b,c)) = \frac{1}{1 + e^{b(-x^c + a)}}, \tag{3}$$

where $x \in \{E(\boldsymbol{\theta}), A(\boldsymbol{\theta}), S_A(\boldsymbol{\theta}), S_E(\boldsymbol{\theta})\}$. $a$ and $b$ translate and scale the values of the predictive components and $c$ spreads out saturated values of the component.

**Fitting the Model with Gradient Descent.** The model combines four components $A(\boldsymbol{\theta})$, $E(\boldsymbol{\theta})$, $S_A(\boldsymbol{\theta})$ and $S_E(\boldsymbol{\theta})$ by taking the sum of their respective values after applying the logistic function $\sigma$:

$$\begin{aligned} f_{\mathbf{w}}(\boldsymbol{\theta}) \\ := \sigma(A(\boldsymbol{\theta}); \mathbf{w}_A) + \sigma(E(\boldsymbol{\theta}); \mathbf{w}_E) + \\ \sigma(S_A(\boldsymbol{\theta}); \mathbf{w}_{SA}) + \sigma(S_E(\boldsymbol{\theta}); \mathbf{w}_{SE}), \end{aligned} \tag{4}$$

where $\mathbf{w}$ represents the parameters of the logistic functions *i.e.,* $\mathbf{w} = (\mathbf{w}_A, \mathbf{w}_E, \mathbf{w}_{SA}, \mathbf{w}_{SE})$. The logistic fitting function is differentiable, and $f_{\mathbf{w}}(\boldsymbol{\theta})$, the linear combination of these logistic functions, is also differentiable. Further, the Pearson correlation coefficient is also differentiable. Therefore we employ gradient descent to fit $\mathbf{w}$ with the Pearson correlation coefficient as the cost function.

## 2.5 Neural Analysis

In search of how OoD generalization and dissemination of orientation invariance emerge in DNN's, we turn to analyze the neurons' activation in the trained networks. We focus on neurons in the penultimate layer of the network, which are attuned to the highest level features in the input stimuli, but reflect a consolidated representation of the entire network for inferring the downstream task (instance classification in our simulations).

In this section, we outline the process by which we quantify several different network invariance metrics. We first formalize the notation for neural activations for single orientations and for sets of orientations. We then define the invariance score (Eq. 6). Finally, we average together many invariance calculations to arrive at the network invariance metric.

We begin by formalizing our approach to neural activations. In Section 2.3 we introduced $\Psi(\boldsymbol{\theta})$, the network's average accuracy at a specific orientation. We can similarly define the neural activation at a specific orientation, though we do so with more granularity. Namely, we introduce $\Phi_i^n(\boldsymbol{\theta})$, which is the average activation of a neuron $n$ from the set of all penultimate-layer neurons $N$ (*i.e.,* $n \in N$) across all images of an object instance $i$ from the set of all object instances $I$ (*i.e.,* $i \in I$) for a given orientation $\boldsymbol{\theta}$. We normalize the activity of each neuron by dividing the activity level of each image by the maximum activity generated by any image. We exclude any neurons with a maximum activation of 0 from further analysis.

Having defined $\Phi$ we note that it is useful to perform analysis not on single orientations only, but sets of orientations. We demonstrated that under our experimental conditions, orientations can be partitioned into coherent subsets — *in-distribution* and OoD orientations. Further, the OoD orientations can be partitioned into *generalizable* orientations, *i.e.,* those OoD orientations that the network can generalize to, and *non-generalizable* orientations. We refer to the *in-distribution*, *generalizable* and *non-generalizable* orientation sets as *InD*, *G* and ¬*G* respectively. The determination of membership of the *generalizable* and *non-generalizable* orientation sets is as follows: We compute 10% of the maximum value of $f_{\mathbf{w}}(\boldsymbol{\theta})$, the predictive model, in the experiment with 40 *fully-seen* instances. All orientations for which $f$ is greater than the 10% threshold are considered *generalizable* otherwise they are considered *non-generalizable*. This threshold for *generalizable* accuracy is intentionally low, as our goal was to capture challenging out-of-distribution cases where the network performs just above chance. This allows us to include orientations that are borderline in terms of

generalization. See Fig. S8 for OoD accuracy partitioned between *generalizable* and *non-generalizable* — this partition captures model behavior well. We can now compute the average activation of a set or orientations. For example, the average activation for a given neuron $n$ and object instance $i$ of the *generalizable* orientations is defined in the following way:

$$\bar{\Phi}_i^n(G) = \frac{1}{|G|} \sum_{\theta \in G} \Phi_i^n(\theta). \tag{5}$$

The same may be computed for *in-distribution* and *non-generalizable* orientations.

To determine how dissemination occurs in the network, we calculate the degree of similarity in a neuron's response to a given instance across different orientations. Specifically, given a neuron $n$ and instance $i$, we calculate the similarity between the neuron's response at an orientation pair $\Phi_i^n(\boldsymbol{\theta}_1)$, and $\Phi_i^n(\boldsymbol{\theta}_2)$, or pair of sets of orientations $\bar{\Phi}_i^n(InD)$, $\bar{\Phi}_i^n(G)$ for example. We use $\delta$, *invariance score*, as the similarity metric, which is defined (based on previous work (Madan et al., 2022)) in the following way:

$$\delta(\bar{\Phi}_i^n(InD), \bar{\Phi}_i^n(G)) = 1 - \left| \frac{\bar{\Phi}_i^n(G) - \bar{\Phi}_i^n(InD)}{\bar{\Phi}_i^n(G) + \bar{\Phi}_i^n(InD)} \right|. \tag{6}$$

We note that under some conditions, $\delta$ reports a high, yet trivial, invariance. Namely, if the response of a neuron is low or zero for both elements of the pair, the denominator approaches zero and the invariance becomes large. However in this case the neuron is not responding to anything — any activity is most likely noise. We therefore calculate a threshold of activity for neural response invariances to be considered to contribute to the generalization capability of the network. Otherwise, these invariances are not integrated into the overall network invariance metric. This threshold ensures that we focus on neurons that are clearly active, reinforcing our interpretation of them as meaningful feature detectors, and it avoids relying on marginal or noisy activations that may not be functionally relevant. The threshold, $\tau$, is the 95th percentile of activity for all neurons across all images. We employ $\tau$ with an indicator function as follows:

$$\mathbf{1}(\bar{\Phi}_i^n(InD), \bar{\Phi}_i^n(G))$$
$$:= \begin{cases} 1 & \text{if } \bar{\Phi}_i^n(InD) \geq \tau \wedge \bar{\Phi}_i^n(G) \geq \tau \\ 0 & \text{otherwise} \end{cases}.$$

Finally, we can compute the overall network *generalizable* and *non-generalizable* invariance scores. To do so, we compute a triple average: an average activation over the set of orientations (Eq. 5) and averaged over the invariance of all neurons and object instances. We say that the *generalizable* invariance score is the invariance between the *in-distribution* orientations and the *generalizable* orientations determined as follows:

$$\frac{1}{L} \sum_{n \in N} \sum_{i \in I} \mathbf{1}(\bar{\Phi}_i^n(InD), \bar{\Phi}_i^n(G)) \cdot \delta(\bar{\Phi}_i^n(InD), \bar{\Phi}_i^n(G)), \tag{7}$$

where $L$ is the quantity of activity pairs above the threshold $\tau$, *i.e.,*

$$L = \sum_{n \in N} \sum_{i \in I} \mathbf{1}(\bar{\Phi}_i^n(InD), \bar{\Phi}_i^n(G)). \tag{8}$$

The definition of the network's *non-generalizable* invariance score is the same, though $\neg G$ replaces $G$.

## 3  Results

**Overview.** In exploration of the generalization capabilities of DNNs to novel orientations, our computational experiments show that the OoD orientation space is divided between *generalizable* and *non-generalizable* orientations, in terms of the networks behavior. We find this division to be governed by a set of rules which determine a partitioning of the orientation space, given a *seed* of orientations for test instances

seen by the DNNs at during training. Among several partitioning rules, we identify a rather intuitive one: small 3D perturbations around seen orientations will be included in the highly-generalizable partition. We find other rules to be more surprising, including that the highly-*generalizable* partition also consists of shape and silhouette preserving rotations, such as *in-plane* (*i.e.,* 2D) rotations and flips along axes of symmetry of the seen *seed* orientations. In order to quantitatively assess this hypothesis, we evaluate the degree of correlation between predicted network behavior induced by these partitioning rules and measured behavior. We find them to be highly correlated under a variety of training regimes. We also explore the DNNs' internal representations and identify neuronal mechanisms that allow for the dissemination of orientation-invariance from familiar objects to novel objects and orientations.

### 3.1 Per-Orientation Accuracy Heatmaps

A detailed inspection of the network's generalization capability for OoD orientations is enabled by introducing per-orientation accuracy heatmaps. Recall from Sec. 2.3 that the continuous space of object orientations, represented by Euler angles, is discretized into *cubelets* (Fig. 3a) – local areas of the rotation space. For each *cubelet* the network's performance is evaluated in terms of the classification accuracy $\Psi(\boldsymbol{\theta})$, where $\boldsymbol{\theta}$ is an orientation of interest. Accuracy heatmaps are 2D projections across a specified dimension of the full accuracy orientation cube (Fig. 3b,c; Section 2.3). These heatmaps reveal a structured pattern of generalization in the form of increased classification accuracy for OoD (*i.e.,* novel) orientations.

For example, Figure 3b shows that for *seed* orientations at the center of the heatmap (red box), the network (in this experiment - ResNet18 (He et al., 2016); see Section 2.2) yields the highest accuracy (brightest *cubelets*) for adjacent orientations around the *seed*, depicting small 3D perturbations of the *seed* orientations. Further inspection of the heatmap reveals other orientations, in this example, brighter *cubelets* forming the figure '8' (stretching the *seed* sideways and along the heatmap's boundaries, enclosing two darker 'holes'), for which the network performs better than for the rest of the OoD orientations. These orientation mainly depict *in-plane* rotations of the *seed* orientations. See also Figure S2, a heatmap where the seed is in the "hole" ($\hat{a}'$ seed). This heatmap yields an inverse of the figure "8" pattern. This heatmap demonstrates a case of "silhouette" generalization, as the airplane is only seen from the front for partially-seen instances, but the model generalizes to views of the airplane from the back.

Figure 3b also shows qualitatively that an increase in number of *fully-seen* instances leads to stronger OoD generalization in the aforementioned orientations (see also Fig. S2). This phenomenon can be reliable quantified. Figure 4a reports the OoD accuracy considering the average classification accuracy across all OoD orientations. It shows an increase in OoD accuracy as data diversity (*i.e.,* the number of *fully-seen* instances) increases, under various conditions, including different *seed* orientations, different image datasets and across datasets. The accuracy heatmaps provide a more detailed means of assessment to the overall average accuracy measure, depicting the generalization patterns and indicating which orientations account for the network increased performance (*i.e.,* Fig. 3b).

The patterns of increased accuracy depict a partitioning of the orientation space, which reappears for various *seed* orientations (Figs. 3c, S1) various sizes of the training set and different object categories (e.g., *Airplane*, *Car*, *Shepard & Metzler (SM)* objects, see Fig. S2), other architectures (see Fig. S3), and training controls (see Fig. S4). Next, we model the patterns of generalization observed in these qualitative results in order to gain more insight.

### 3.2 Modeling Generalization Patterns

We formulate a model of partitioning rules, which can be used to predict the OoD generalization patterns of the network, given a *seed* of *in-distribution* orientations. Recall from Sec. 2.4, that the model, denoted by $f_{\mathbf{w}}(\boldsymbol{\theta})$, has three components: $A(\boldsymbol{\theta})$, which captures small angle rotations around $\boldsymbol{\theta}$; $E(\boldsymbol{\theta})$, which captures in-plane (2D) rotations; $S(\boldsymbol{\theta})$, which captures object silhouette projections at the orientation $\boldsymbol{\theta}$ (see details in section 2.4). We evaluate the model's performance by measuring the Pearson correlation coefficient $\rho$ between the accuracy of the networks as measured in our experiments and as predicted by the model, *i.e.,* $\rho(\Psi(\boldsymbol{\theta}), f_{\mathbf{w}}(\boldsymbol{\theta}))$.

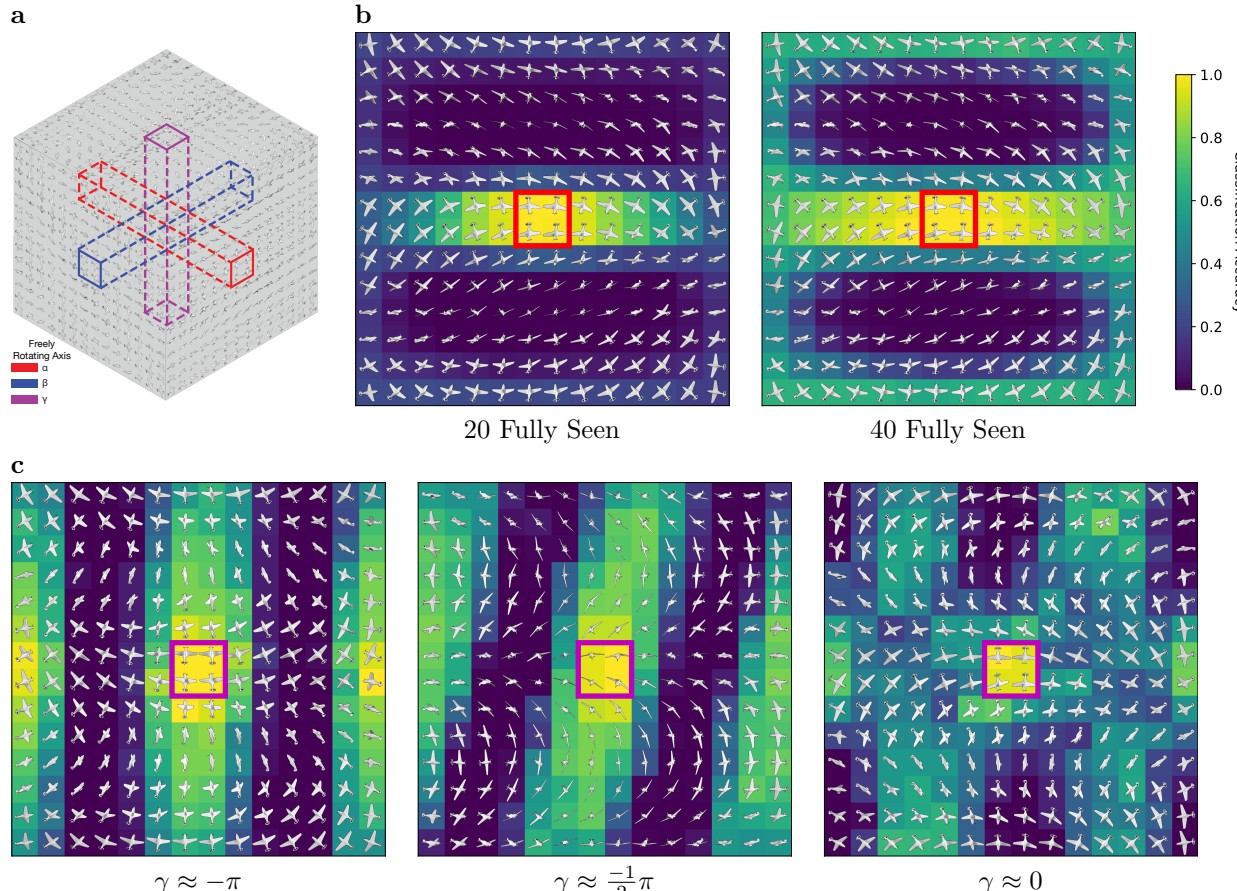

Figure 3: **Observed generalization patterns in per-orientation accuracy heatmaps.** When trained with a combination of *fully-seen* instances and *partially-seen* instances, DNNs demonstrate the ability to generalize outside of their training distribution. Generalization behaviors are demonstrated measuring per-orientation accuracy. **(a)** All orientations can be described by three Euler axes ($\alpha, \beta, \gamma,$) and rotations are periodic around these axes. These properties allow for the visualization of all possible orientations with an orientation cube, shown here. The orientations contained within the colored rectangular prisms are the *seed* orientations — those orientations of the *partially-seen* instances included in training (*i.e.,* are *in-distribution*). The *seed* orientations differ depending on the experiment. All other orientations are OoD. **(b)** Increased network generalization for OoD orientations, with increased instance diversity (*i.e.,* number of *fully-seen.*) Each cell in the heatmap is the average classification accuracy of the network for a given value of $\beta$ and $\gamma$, across all values of $\alpha$. Chance level is 0.02 (2%), single repetition, *seed* is $\hat{\alpha}$. **(c)** Different *in-distribution* parameters (in this case, $\hat{\gamma}$ *seed*) affect generalization behaviors. Single repetition.

Figure 4b-top shows the predictive power of the model and its components in experiment with different *seed* orientations and several object categories. We conducted a large series of experiments under various settings, including different *seed* orientation distributions, various amounts of training examples, object categories with different levels of symmetry (Fig. S5). In all experiments our model highly predicts the network's behavior, indicating that indeed the networks generalization patterns for OoD orientations follow the model's partitioning rules. This is true even across categories, when the *seed* is taken from one category (*e.g., SM*) and the *fully-seen* instances are taken from another (*e.g., Airplane*).

We now analyze the contribution of each component of the model to predicting per-orientation OoD accuracy. The model's component $A(\boldsymbol{\theta})$ ('small-angle' rotations), is the best predictor for the network's OoD behaviour, for highly articulated objects such as the *SM* objects. The model's component $E(\boldsymbol{\theta})$ ('in-plane' rotations), is a better predictor for non-articulated objects with inherent symmetries. Finally, the model's component

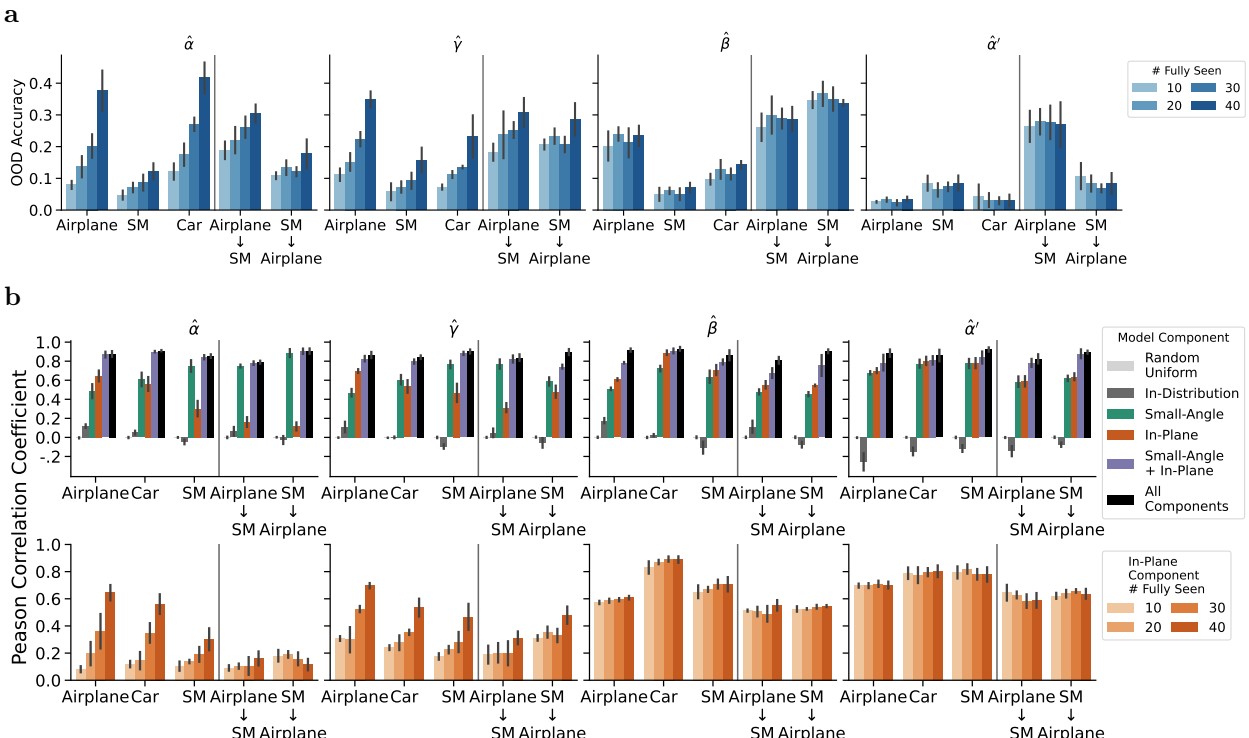

Figure 4: **Modeling generalization patterns for OoD orientations.** The bar plots show several trends related to DNN OoD classification patterns. The trends are measured under the various controls, including *in-distribution* orientations conditions ($\hat{\alpha}, \hat{\gamma}, \hat{\beta}, \hat{\alpha}'$) and object category, which is either a single object as in Airplane, SM, Car, or transfer across two categories, when the *fully-seen* instances are of a different category than the *partially-seen* instances as in Airplane $\rightarrow$ SM and vice versa. These transfer cases are visually separated from the other cases. **(a)** Network generalization for OoD orientations increases with increasing number of fully seen (blue shading.) This trend holds across in-distribution orientations (*seed*) and object category conditions — including when *fully-seen* and *partially-seen* are the same category, and when they differ (*i.e.,* Airplane $\rightarrow$ SM). **(b)** Top: We introduce a predictive model for OoD orientation generalization (black — "All Components") which is highly predictive of experimental results, with greater than 0.8 Pearson Correlation Coefficient for all experimental controls. (Results are shown for experiments with 40 *fully-seen* instances.) Null hypothesis predictive models, including "Random Uniform" and "In-Distribution," have very low correlation coefficients. We also ablate our predictive model, including only some sub-components, like only-"Small Angle", only-"In-Plane" or only-"Small Angle + In-Plane." These ablated models have lower correlation coefficients than "All Components," and vary in relation to one another depending on the experimental condition. Bottom: We isolate the predictive power of the only-"In-Plane" component for all experiments with a range of number of *fully-seen*. The increasing predictive power of the "In-Plane" component correlates with increasing OoD accuracy as the number of *fully-seen* instances increases. This suggests that generalization to "In-Plane" orientations drives OoD accuracy.

$S(\boldsymbol{\theta})$ is predictive of network behavior under special circumstances, when the silhouette of an object appears similar both *in-distribution* and OoD, but this component does not have much of an effect on overall model performance. Further analysis of 'in-plane' component in Fig. 4b-bottom illustrates how generalization to 'in-plane' rotations emerges with the increase in data diversity. Thus, the increase in OoD generalization is primarily to 'in-plane' rotations. On the other hand, the 'small-angle' and 'all-components' models are negatively correlated and uncorrelated with the number of *fully-seen*, respectively. See Fig. S7 for further details, and also Fig. S5 for the same analysis with various training controls and other architectures.

### 3.3 Individual Unit Neuronal Analysis

Figure 5a illustrates activation of individual neurons for stimuli of *fully-seen* and *partially-seen* instances. Each group of images depict input stimuli of a particular instance for which a particular neuron has the highest activation, along with the neuron's per-orientation activation heatmap for the instance. The patterns seen in the neurons' activation heatmaps resemble the partitioning patterns of the accuracy heatmaps shown in Figure 3b. Some neurons exhibit similar activation patterns for both *fully-seen* and *partially-seen* instances, while others do not.

A quantifiable measure of these neuronal responses can help with understanding how generalization occurs in the network – particularly generalization to OoD orientations of *partially-seen* instances, where generalization must stem only from *seed* orientations seen during training. Previous neural analysis approaches, like t-SNE, fail to capture the shared representations between *fully-seen* and *partially-seen* instances, and therefore cannot be used to advance hypotheses related to dissemination (see Fig. S9).

Recall from Section 2.5, we define an activation invariance score in the range $[0, 1]$ (Eq. 6) between sets of orientations, in particular between the *seed* orientations and OoD *generalizable* orientations or *non-generalizable* orientations. The invariance score yields higher values when a neuron fires for both sets of orientations, and lower values when it fires only for one set. See Sec 2.5 for details on how to operationalize partitioning the OoD orientations into *generalizable* and *non-generalizable* orientations, and see Fig. S8 for results that demonstrate that OoD accuracy is well captured by this partitioning.

We now evaluate whether OoD generalization is predicted by invariance. If that is the case, this would provide evidence that invariance underlies OoD generalization. Thus, we analyze whether the network accuracy is explained by the emergence of invariant representations. Figure 5b depicts a scatter plot of the invariance score against the classification accuracy. Each dot represents an experiment (a tuple of number *fully-seen*, object and architecture type) and the coloring indicates the respective instance set (*fully-seen* or *partially-seen*) and orientation set (*generalizable* or *non-generalizable*). There is a clear correlation between increasing levels of classification accuracy and increasing invariance score for the *partially-seen* instances. Furthermore, the plot shows a clear partition between *generalizable* and *non-generalizable* orientations with respect to the invariance score, where significantly higher invariance scores are measured for the *generalizable* orientations.

Note that for *fully-seen* instances (Fig. 5b gray dots), all orientations are *in-distribution*, including the *seed*, *generalizable* and *non-generalizable* (data comes from the validation set). Since all orientations are *in-distribution*, the network achieves accuracy at ceiling levels regardless of the neuronal invariance score. *i.e.,* these orientations fall within the training distribution and the network has learned to associate them with their corresponding object instances. Nevertheless, the *fully-seen* instances exhibit the same invariance partitioning between *generalizable* and *non-generalizable* orientations as the *partially-seen* instances.

These results provide evidence that OoD generalization is driven by emergent invariant representation. However, the emergence of *partially-seen* invariance in *generalizable* orientations is intriguing, as this invariance can not be directly learned since *partially-seen* instances are only seen in *seed* orientations during training. We hypothesize that this invariance is disseminated from the invariance that develops for *fully-seen* instances. To investigate if this may be the case, we analyze the relationship between *partially-seen* and *fully-seen* invariance. Figure 5c depicts a direct comparison between the invariance score of the *fully-seen* and *partially-seen* sets for both the *generalizable* and *non-generalizable* orientations. The partition between *generalizable* and *non-generalizable* orientations is exhibited again — the *non-generalizable* invariances are in the bottom left corner, while the *generalizable* invariances are in the top right corner. Each point in this plot represents the joint invariance of *fully-seen* and *partially-seen* instances at a given orientation. The plot shows a tight correlation between the invariance scores of the *fully-seen* and *partially-seen* instances, as most of the points lie within a band roughly 0.1 units away from the line of parity, $x = y$. This correlation suggests that *partially-seen* invariance emerges due to development of *fully-seen* invariance. See also Fig. S6 for the same analysis with other training controls.

Taken all together, the correlations between *fully-seen* invariance and *partially-seen* invariance, and between *partially-seen* invariance and generalization behavior, provide substantial evidence that OoD generalization

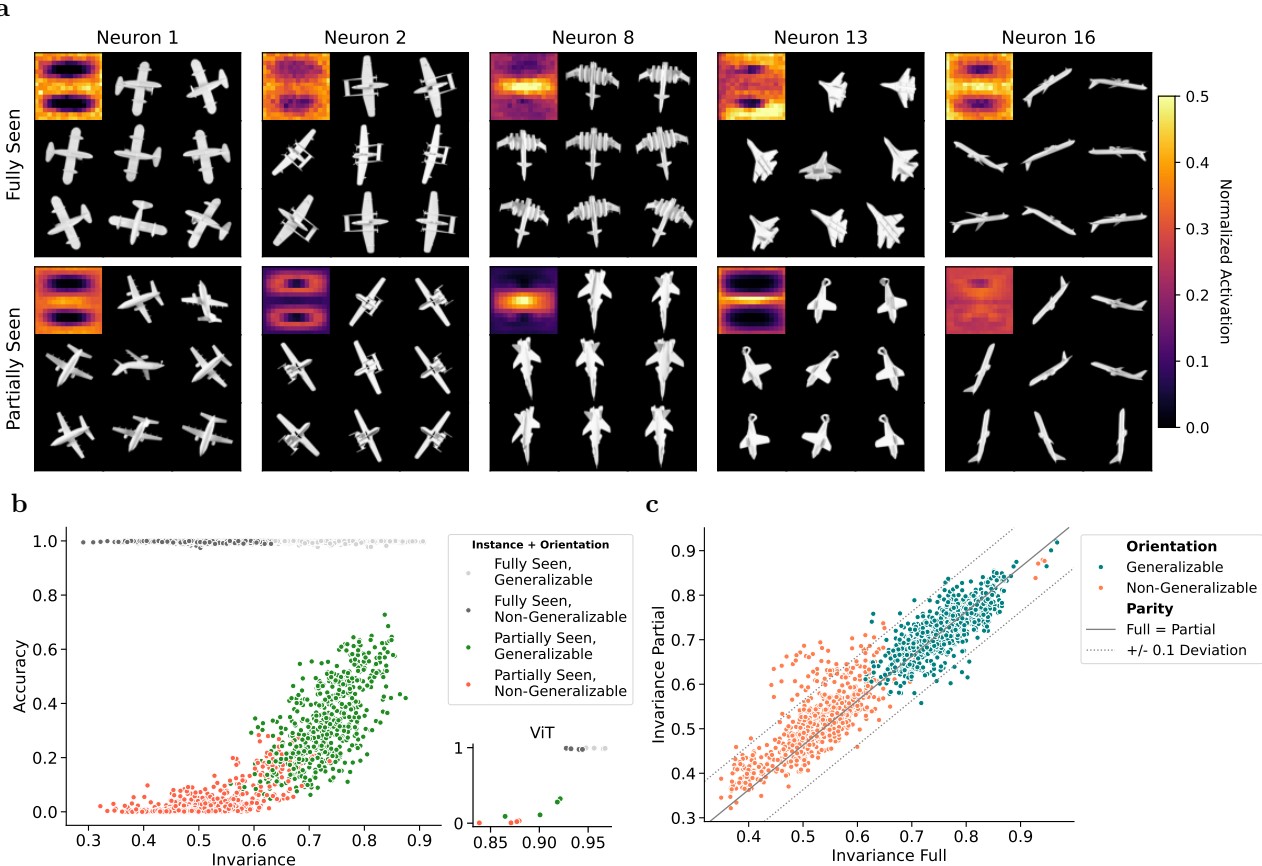

Figure 5: **Neuronal analysis, Invariance and Dissemination. (a)** An intuitive visualization of neural activity. Each square is the response of a single neuron to the airplane instance that most highly activates, it portrayed in two ways: 1) the top-8 images that most highly activate the neuron (in no particular order), 2) the heatmap of the per-orientation normalized neural activity for the airplane instance. Neurons tend to exhibit patterns of activation related to the patterns of generalization behavior (Figs. 3b for example,) and are invariant to a range of orientations that respect the partitioning of OoD orientations. Comparing the neural responses in each column demonstrates that the patterns of activation are similar between the *fully-seen* instance that most highly activates the neuron and the *partially-seen* instance that most highly activates it due to shared visual, part and semantic features between these instances. Several randomly sampled penultimate layer neurons, arranged into columns, demonstrate that these findings apply to many neurons. **(b)** Each experiment is portrayed by four dots, one for each of the invariance scores (Eq. 6), depicted with different colors. Each experiment is a DNN trained on a specific combination *seed*, object category and proportion of *fully-seen*, see Sec. 2. Averaging the activations in the partitioned regions (*seed*, *generalizable*, *non-generalizable*) and computing the invariances (defined here: Eq.6) between *seed* and OoD regions captures overall generalization in the network. Plotting the generalization metrics against accuracy for those regions demonstrates a clear correlation between increasing invariance and increasing OoD classification accuracy (*i.e., partially-seen* instances in OoD orientations.) Visual Transformer results are placed separately to highlight that they follow the same trend, though their invariance is scaled higher. **(c)** Plotting *fully-seen* invariance against *partially-seen* invariance for the same experiment also yields a tight correlation, suggesting that dissemination of invariance from *fully-seen* to *partially-seen* instances enables increasing generalization in OoD orientations of *partially-seen* instances.

is driven by dissemination of internal network invariances from *fully-seen* to *partially-seen* instances through shared features.

# 4 Conclusions

In this work we analyze the generalization behaviors of DNN's on rendered images from all orientations and observe dissemination of orientation-invariance for orientations that appear like 2D rotations (*in-plane*) of *in-distribution* orientations. For *non-generalizable* orientations, the network has not developed orientation-invariance with respect to the *seed* orientations (demonstrated by the lower invariance score in our results). Our results support the hypothesis that the network disseminates orientation-invariance of *fully-seen* instances to *partially-seen* instances using brain-like mechanism similar to those reported by (Logothetis & Sheinberg, 1996; Poggio & Anselmi, 2016). Neurons are feature detectors, and during training neurons are tuned to detect the features of *fully-seen* objects at multiple orientations — *i.e.,* the neurons become selective to the feature, but invariant to the orientation. Some features that neurons are tuned to are shared between *fully-seen* and *partially-seen* instances (Fig.5a). Therefore the invariance that develops for features of *fully-seen* instances are gained "for free" for *partially-seen* instances in the same orientations. Our results provide a quantitative assessment of this hypothesis and elucidate the intricate neural processes involved in object recognition, underscoring the critical role of individual neuron, feature-based representations for OoD object recognition.

**Limitations and future work.** This study reveals discernible patterns in the successes and failures of DNNs across diverse orientations which can be effectively characterized and explained through the analysis of neural activity. This underscores the potential for more comprehensive analyses of DNNs that transcend the conventional approach of solely focusing on average accuracy. This study was limited to a supervised classification setting, and to fairly small DNNs. Nonetheless a promising research avenue is to apply the analytical methods introduced in this paper to other newer models.

A key question arising from our results is to explain why DNNs disseminate orientation-invariance only to *in-plane* orientations. All object instances are distinguishable at all orientations, as evidenced by the high *in-distribution* accuracy achieved by the DNNs. Therefore the lack of orientation-invariance for such *non-generalizable* orientations is an outcome of the DNN's learning process. We speculate that this may be because orientations that are not *in-plane* are affected by self-occlusion, which poses a particular challenge for DNNs (Michalkiewicz et al., 2024).

Furthermore, various efforts have been made to enhance DNNs' generalization capabilities to OoD orientations including leveraging preconceived components for DNNs, such as 3D models of objects (Angtian et al., 2021), sophisticated sensing approaches like omnidirectional imaging (Cohen et al., 2018) or novel architectures like Light Field Networks (O'Connell et al., 2025). Other works, including by Cohen and Welling (Group Equivariant CNN's (Cohen & Welling, 2016) and Steerable CNN's (Cohen & Welling, 2017)) induce invariance by construction with modifications to the CNN architecture. These works generalize the translation inductive bias inherent in CNN's to broader mathematical groups, including rotations. However, these approaches focus on architectural improvements that augment the underlying capabilities of these networks, but don't investigate whether the vanilla architectures exhibit emergent orientation generalization. Instead, novel approaches that extend the emergent orientation-invariance inherent within networks might allow for further gains of OoD generalization. Biological agents may overcome the difficulties associated with recognizing OoD orientations by leveraging the temporal dimension to associate orientations and learn invariant representations (Ruff, 1982; Johnson & Aslin, 1996; Ratan Murty & Arun, 2015). The mechanisms that utilize temporal association may hold fundamental significance, given that they have access to a plentiful source of training data that does not rely on external guidance and task specific labels. This data is readily available prior to any visual task and has the potential to contribute to the emergence of orientation-invariant representations beyond *in-plane* orientations.

Previous studies have extensively compared the behavioural and electrophysiological aspects of brains and DNNs (Yamins et al., 2014; Yamins & DiCarlo, 2016). However, a direct comparison between these systems alone has limitations in providing insights into the underlying mechanisms of object recognition in DNNs. This is due to the possibility that while certain fundamental mechanisms may be shared across these systems, the manifestation of these fundamental mechanisms can differ at the behavioral and electrophysiological levels. Our study has provided compelling evidence of brain-like neural mechanisms in DNNs that facilitate

object recognition in novel orientations, even though these mechanisms are manifested differently than in biological systems. For instance, while humans and primates can recognize objects in orientations that are not simply 2D rotations, this capability is not fully replicated in DNNs. Thus, we can conclude that the neural mechanisms that have been observed to govern recognition in biological systems largely apply to DNNs, albeit with distinct manifestations across these systems. It will be interesting to follow this line of investigation across biological and artificial systems to envision a general theory to explain emergent mechanisms in both brains and machines.

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

## Supporting Information

- S1: Accuracy heatmaps: alternative *seed* orientations.

- S2: Accuracy heatmaps: effect of data diversity - alternative object categories.

- S3: Accuracy heatmaps: alternative backbone architectures.

- S4: Accuracy heatmaps: alterative training conditions - pretraining and augmentation.

- S5: Alternative architectures and ablation studies: modeling generalization patterns.

- S6: Ablation studies: invariance and dissemination.

- S7: Other predictive model components, by number of *fully-seen*

- S8: OoD accuracy, split between *generalizable* and *non-generalizable* orientations

- S9: tSNE analysis on the penultimate layer of a representative experiment.

**a**

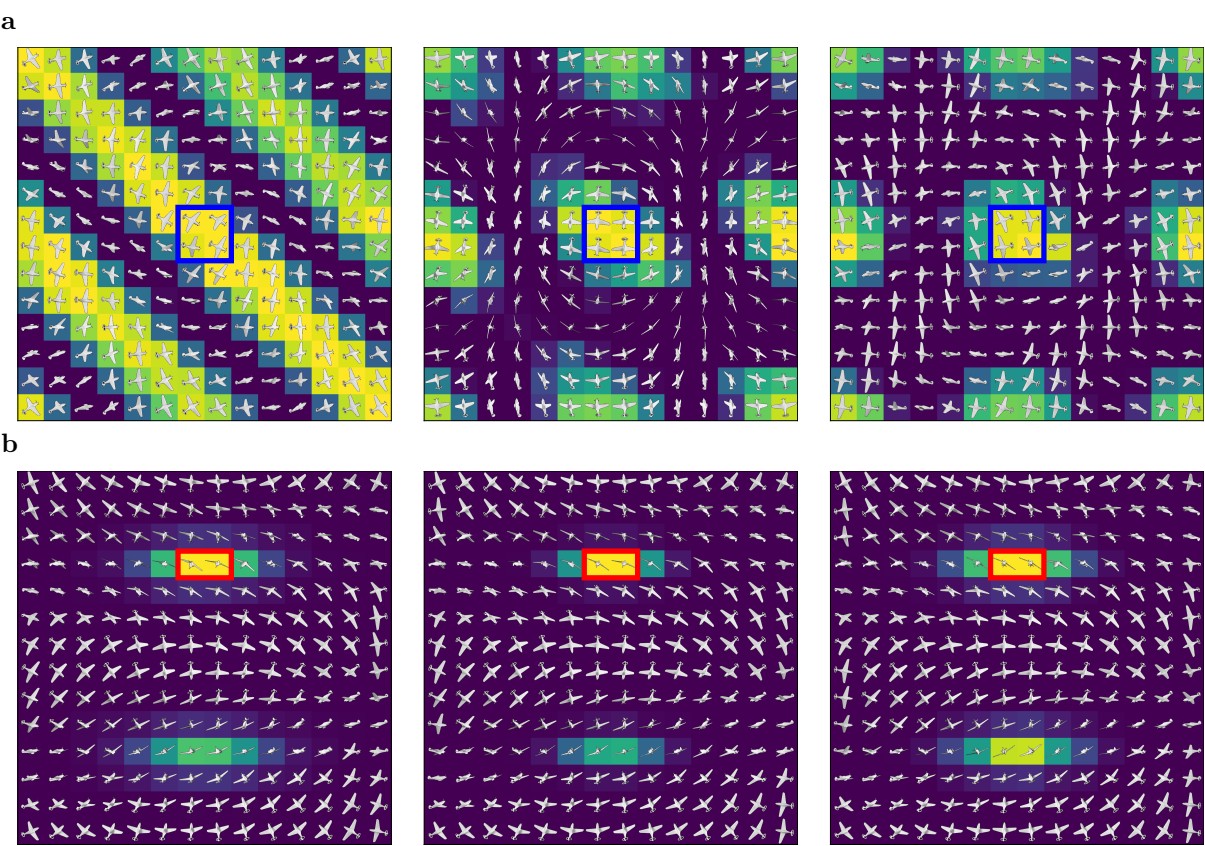

**b**

Figure S1: **Accuracy heatmaps: alternative *seed* orientations.** (**a**) $\hat{\beta}$ *seed*. (**b**) $\hat{\alpha}'$ *seed*.

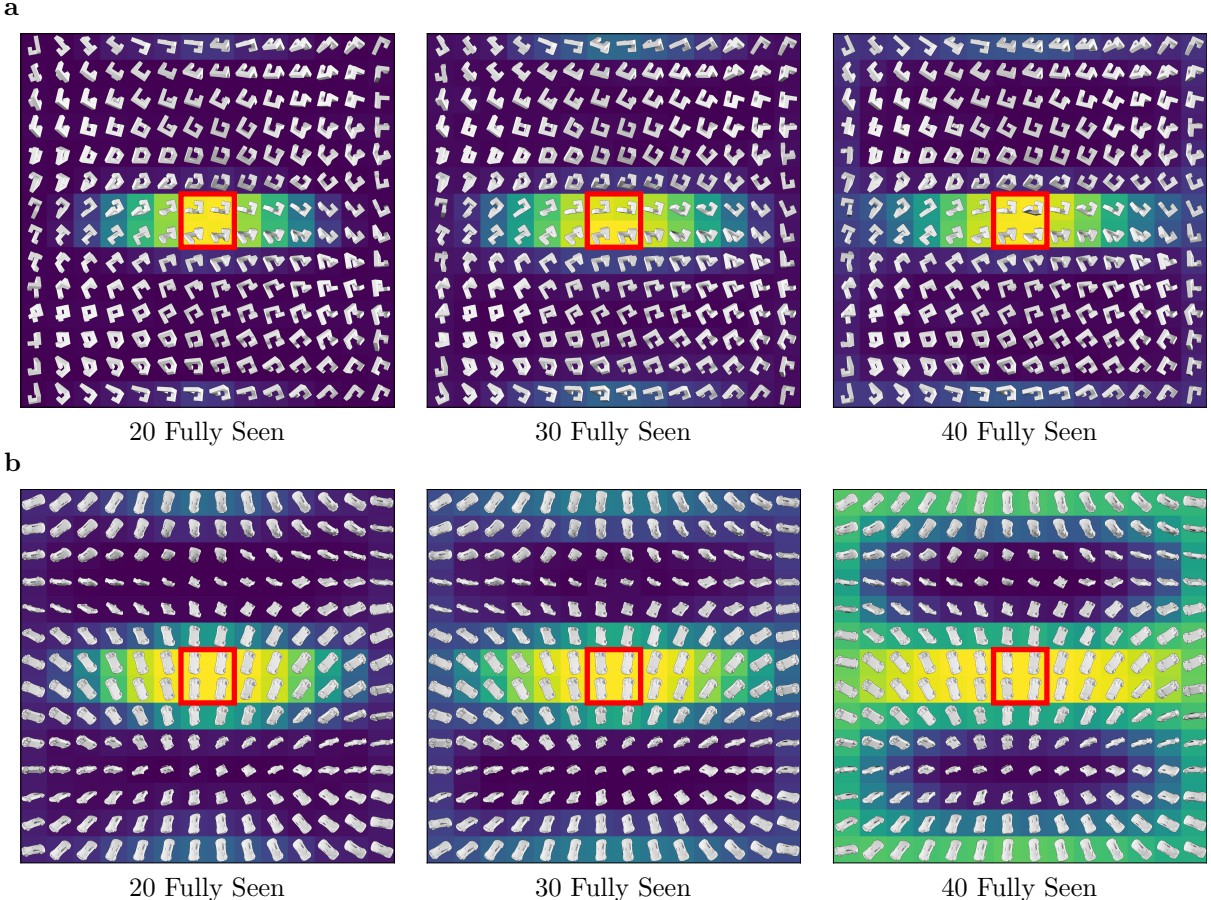

Figure S2: **Accuracy heatmaps: effect of data diversity - alternative object categories.** Increasing number of *fully-seen* instances, with different object classes. (**a**) Shepard-Metzler Objects. (**b**) Cars.

**a**

**b**

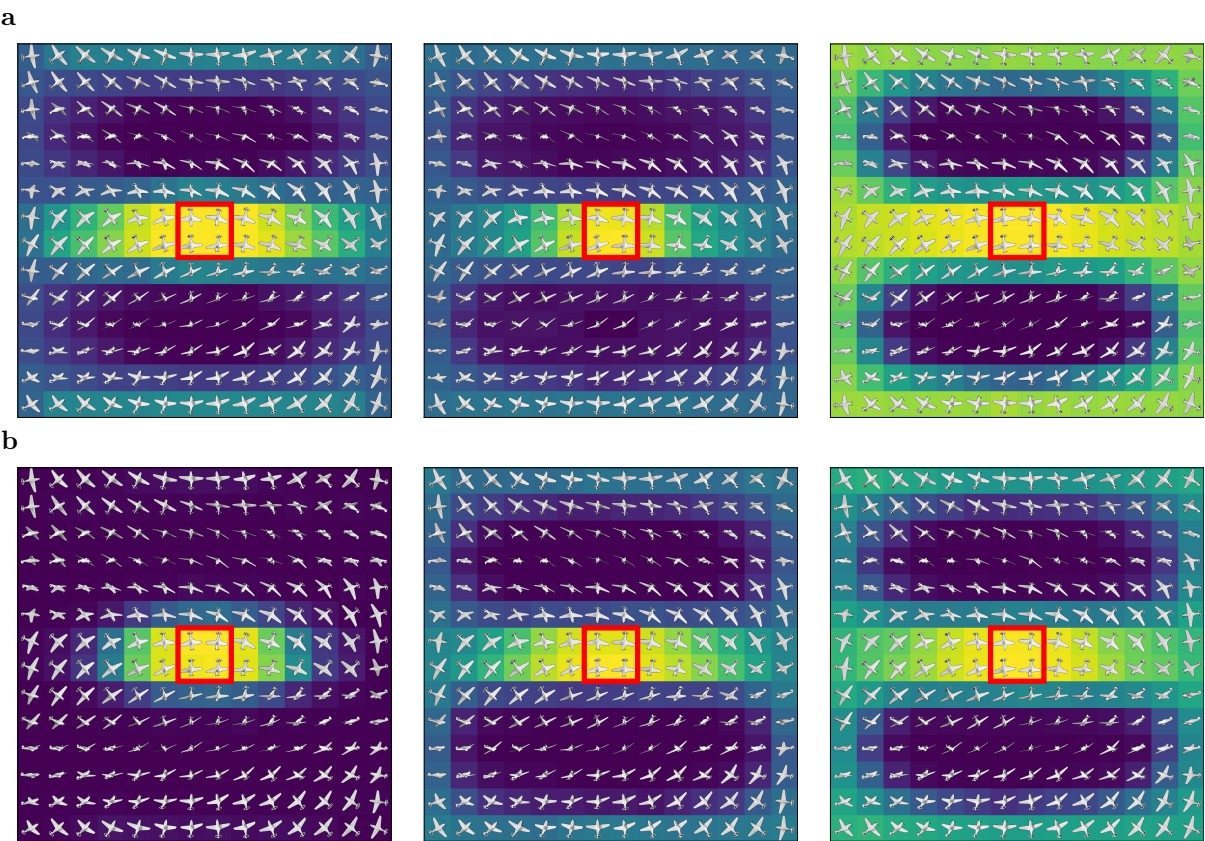

Figure S3: **Accuracy heatmaps: alternative backbone architectures.** (In place of ResNet-18): (**a**) DenseNet121. (**b**) CORnet-S.

**a**

**b**

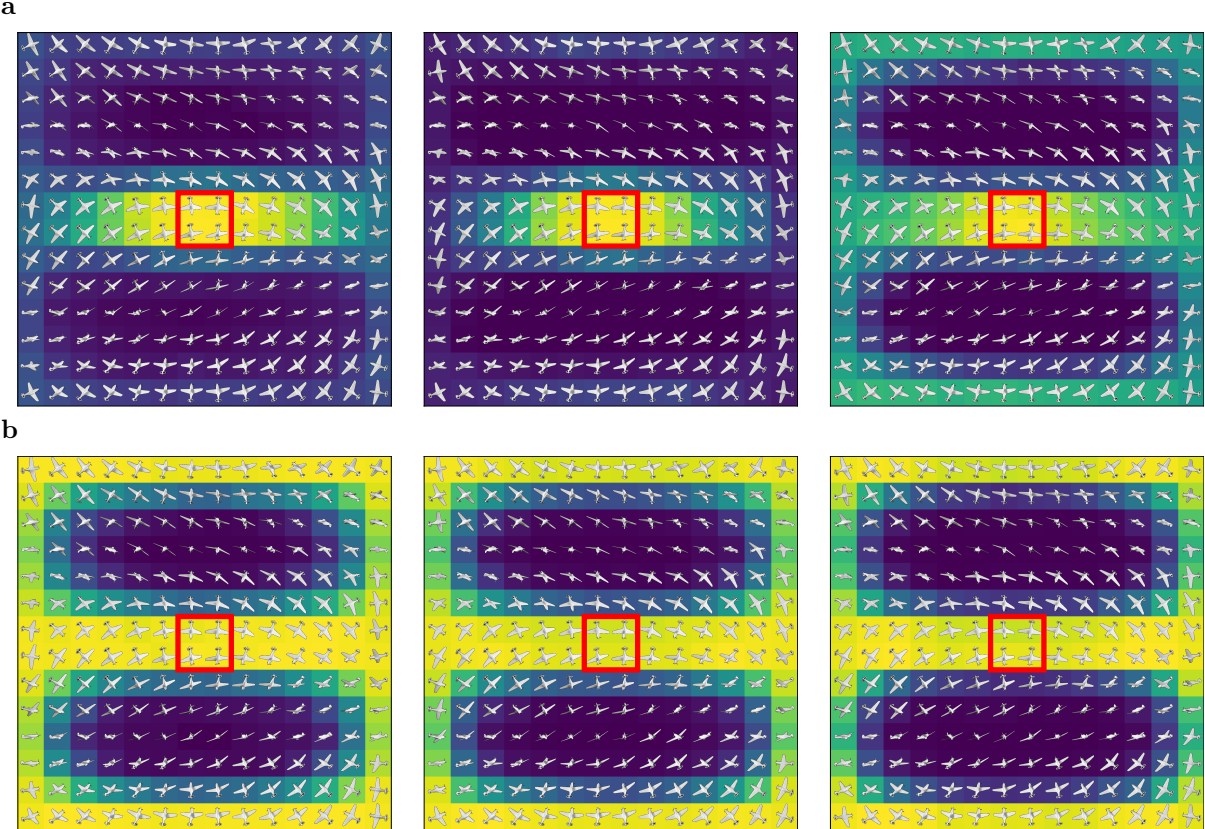

Figure S4: **Accuracy heatmaps: alterative training conditions - pretraining and augmentation.** (**a**) ResNet-18 pretrained on ImageNet Russakovsky et al. (2015), finetuned on our learning paradigm with airplanes. Network behavior isn't meaingfully altered. (**b**) All data (both from *fully-seen* and *partially-seen* instances) were augmented with random 2D image rotations. This effectively expands the *in-distribution* set to include all *generalizable* orientations. This results in *generalizable* orientations with high accuracy.

**Ablation Studies** In order to better understand how certain experimental design choices may effect our results, we conducted several ablations with ResNet-18. These ablations include:

- **Pretrained:** On ImageNet Russakovsky et al. (2015), finetuned on our learning paradigm with airplanes.

- **Augmented:** All data (both from *fully-seen* and *partially-seen* instances) were augmented with random 2D image rotations. This effectively expands the *in-distribution* set to include all *generalizable* orientations.

- **Half-Data:** 50% of the samples from the full experiment for each subset (*i.e.,* each instance in the appropriate orientations.)

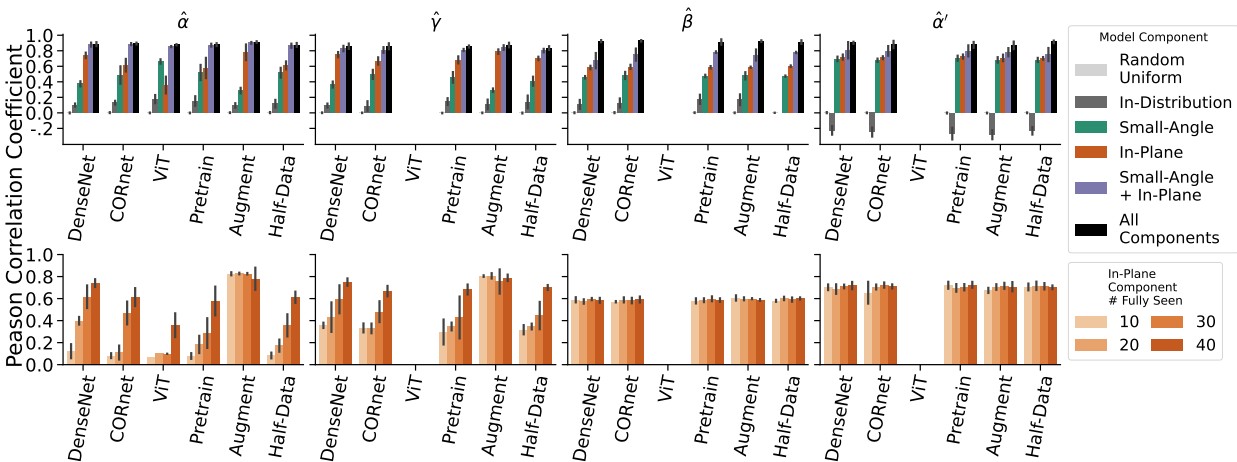

Figure S5: **Alternative architectures and ablation studies: modeling generalization patterns.** The same analysis as Fig. 4b is applied to the other architectures and the ablation controls enumerated above.

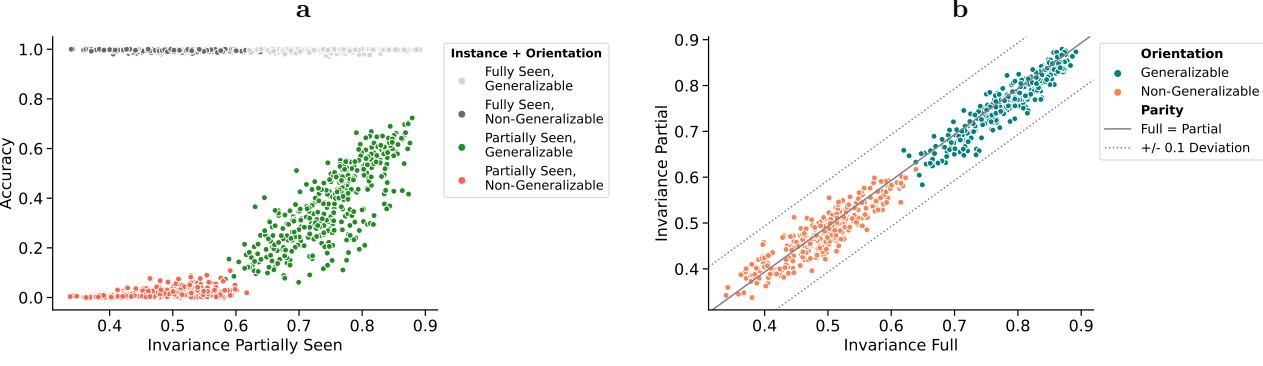

Figure S6: **Ablation studies: invariance and dissemination.** The same analysis as Figs. 5b,c is applied to the ablations controls enumerated above.

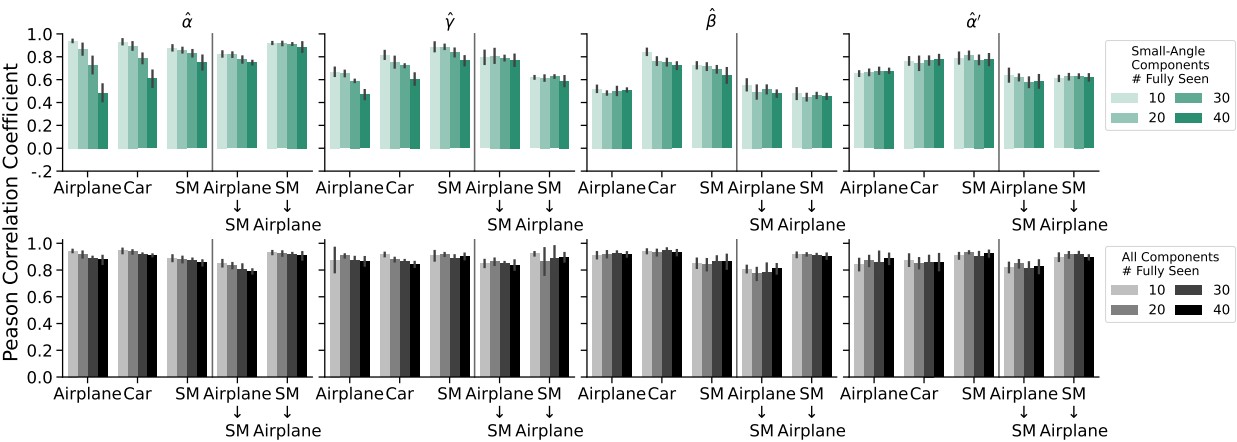

Figure S7: **Other predictive model components, by number of *fully-seen*** In Fig. 4b we report the correlation coefficient for the In-Plane Component for the number of *fully-seen* instances. We chose this component specifically, as it is well correlated with the number of *fully-seen* instances, *i.e.,* most of the OoD generalization is to In-Plane orientations. In this figure we show the same plots but for the Small-Angle component and the All-Components models. The former is negatively correlated with number of *fully-seen*, *i.e.,* it predicts OoD generalization with few *fully-seen*, but not with increasing *fully-seen*. The latter has no change in correlation, as it a linear combination of the Small-Angle and In-Plane components, and changes the weigting between the two to maximize correlation at each number of *fully-seen*.

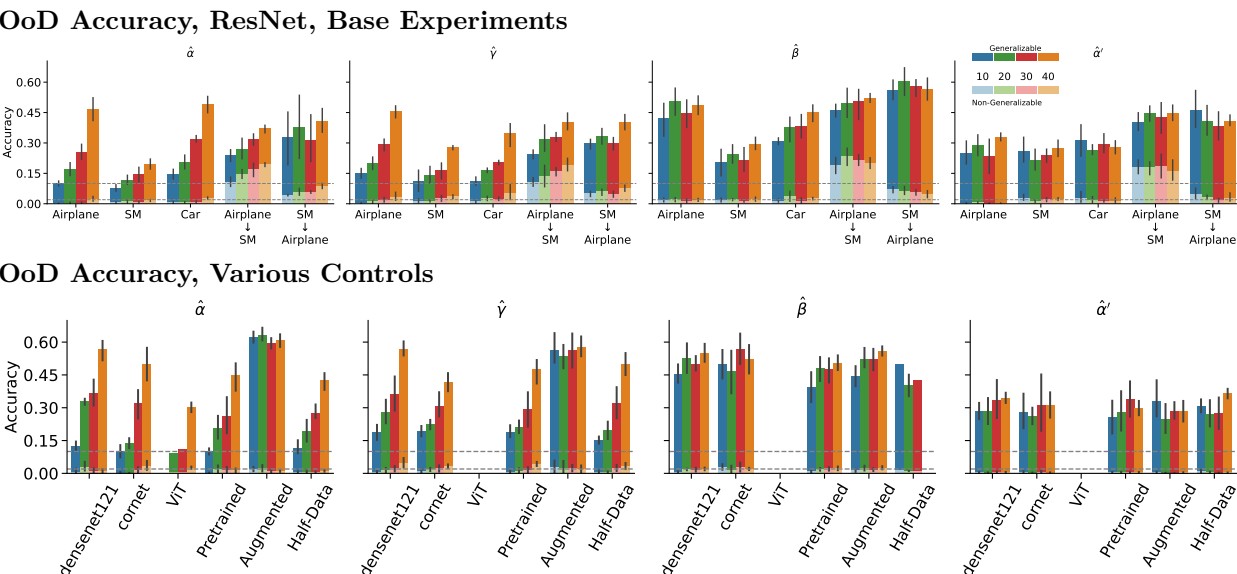

Figure S8: **OoD accuracy, split between *generalizable* and *non-generalizable* orientations.** In Fig. 4a we report the average accuracy across all OoD orientations. As we note, however, accuracy behavior is differentiated between *generalizable* and *non-generalizable* orientations. Here we report the average accuracy for these two orientation groups. Gray horizontal lines indicate chance performance of 2% and 10% (the latter relevant in the case where *fully-seen* and *partially-seen* instances are of two different classes.) *Generalizable* accuracy is always greater than *non-generalizable* accuracy. The former is always well above chance, while the latter is below or at chance level. **(a)** The *generalizable* and *non-generalizable* average accuracy for the same set of experiments presented in Fig. 4a. **(b)** The average accuracies for several other conditions. These other conditions are alternative architectures and training controls (explained above Figs. S5 S6).

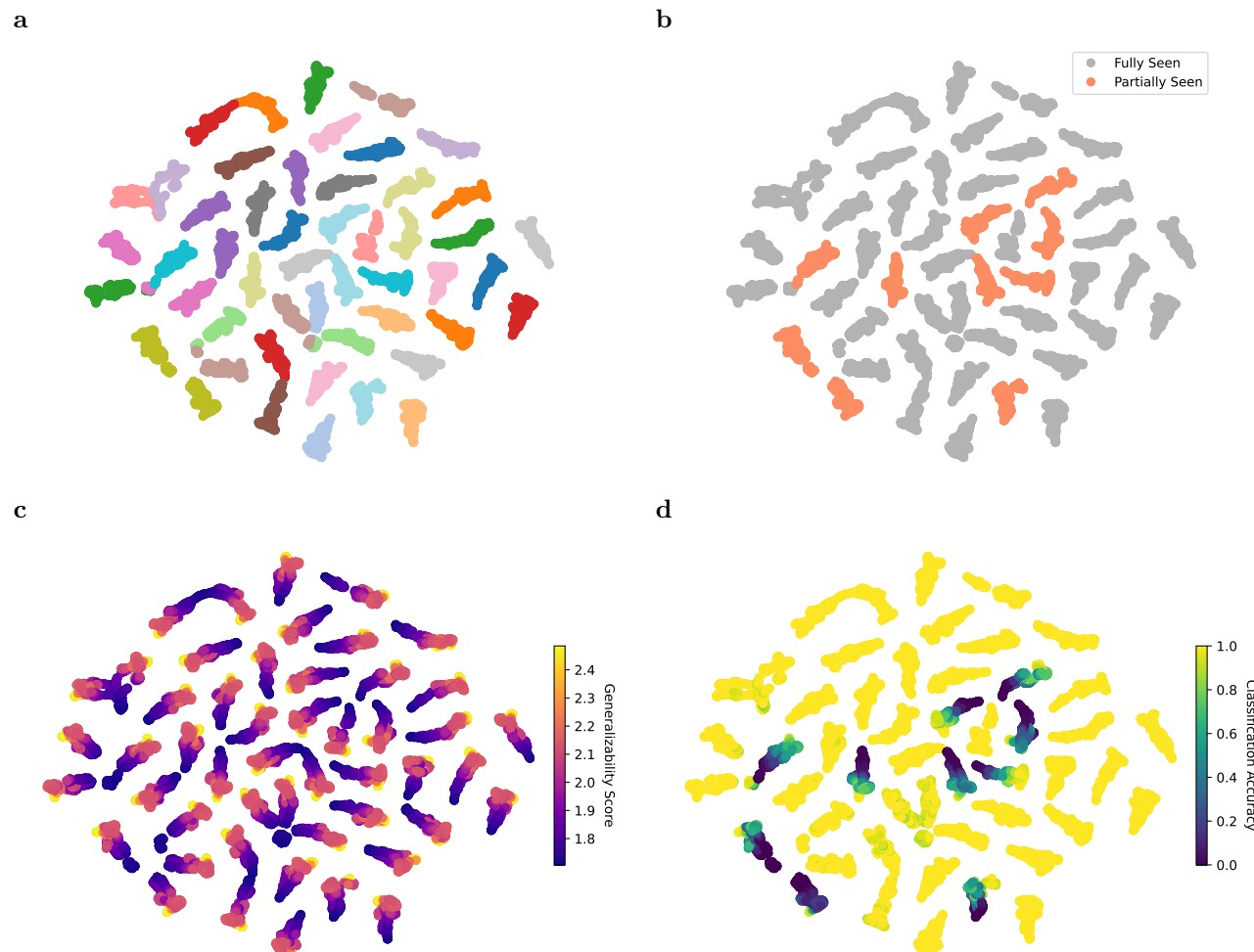

Figure S9: **tSNE analysis on the penultimate layer of a representative experiment.** The 512 dimension activation vectors in the penultimate layer for each instance and orientation are recorded. We employ tSNE to reduce these 512 dimensions down to two dimensions. **(a)** Object instances are colored (semi-) uniquely. (20 colors are distributed to 50 instances due to the limits of choosing many perceptually different colors.) For the most part, instances cluster together without much overlap between clusters of different instances. This indicates that representations of instances are separable, and that the task is solved by DNN. **(b)** Each point is colored based on whether the instance it represents is *fully-seen* or *partially-seen*. *Partially-seen* clusters are independent of other clusters, both *fully-seen* and *partially-seen*. It is therefore difficult to determine the range of behaviors for *partially-seen* instances — namely, why certain OoD orientations are *generalizable*, while others are not. **(c)** Points are colored with the degree of generalizabilty, as predicted by the predictive model of DNN generalization behavior. Note that points within each cluster are ordered — they are arranged such that *generalizable* orientations are far from *non-generalizable* orientations with a smooth transition between them. **(d)** Points are colored with the classification accuracy of the network (for the given instance and orientation.) While *fully-seen* instances have near 100% accuracy across all orientations, *partially-seen* show differentiation in accuracy between *generalizable* and *non-generalizable* orientations.

