# OpenReview forum: "Emergent Neural Network Mechanisms for Generalization to Objects in Novel Orientations"
_TMLR — Accepted by TMLR_

### Review · Reviewer_viQw · 2025-06-19

**Summary Of Contributions:**

This paper studies the ability of deep neural networks (DNNs) to generalize to objects in orientations that are not seen during training. This paper studies this question through the following experimental setup: during training, objects of a particular class are either seen in all orientations (denoted fully-seen), whereas other objects are only shown in a subset of the orientations (denoted partially-seen). Generalization is measured by test-time performance on the unseen orientations of objects partially-seen during training. Training was presumably done in the supervised fine-grained image/instance classification setting, and the metric used to evaluate generalization is classification accuracy. The paper finds that DNNs generalize out-of-distribution to shape and silhouette preserving orientations as well as orientations that are small perturbations of orientations seen during training. The paper also examines generalization at the neuron level and presents a logistic regression model that quantitatively evaluates the paper’s hypothesis.

**Audience:**

Yes

**Claims And Evidence:**

No

**Requested Changes:**

1. Clarify the experimental settings, which experimental settings were used to generate the figures/plots, and undefined notations
2. Reorganize the paper to put Methods before Results
3. Address limitations of this work, e.g. limitation to small networks sizes, limitation to only supervised training, limitation to only examining classification. I think this can be addressed by repeating the experiments on larger networks and self-supervised networks as well as another task.
4. Address potential overclaims (listed above)
5. Explain some additional hyperparameter choices
6. Show more clearly the results of multiple runs

**Strengths And Weaknesses:**

[Strengths]
This paper studies the generalization of DNNs to novel orientations unseen during training, which should be important to machine learning researchers working with 3D data.

[Weaknesses]

[Clarity]
I would prefer if the experimental setup were made more explicit. In particular, it would be good to clarify the training task, e.g. is it supervised? How are $\alpha, \beta, \gamma$ sampled, and how are the ranges for sampling chosen? At test-time, are the in-distribution images tested exactly those used during training, or are they just drawn from the same distribution as the training distribution? Which plots correspond to which sampling settings?

I also think the organization of this paper makes the paper hard to understand. I would prefer if the Methods were presented before the Results, which seems to be the standard format for most machine learning papers.

In Figure 3, the $\alpha, \beta, \gamma, \alpha’$ labels are not explained, and it's not clear which architecture(s) this plot was generated from. Plots (e.g. Figure 4b, 4c) are not labeled with what dataset and model the plot was generated from.

For models where there is not one set configuration (e.g. ViT, which frequently comes in tiny, small, base, giant, etc. sizes), it would be good if the authors could report the number of parameters.

[Limitations of the experimental setup]
This paper only examines small networks (e.g. ResNet-18), (presumably) trained to do supervised instance image classification.The focus of this paper on supervised training is a major limitation of this paper, as self-supervised models (many of the large models used today) behave differently than supervised models. I also think that ideally, tasks other than classification should be examined, as classification may not require a more limited understanding than, say, segmentation, and may have a different pattern of generlization to OOD orientations. It’s not clear that we can draw conclusions about generalization in such a scenario.

[Potential overclaims]
As discussed above, I think the implication that this work may have ramifications for Qwen2-VL and Llama 3-V (last paragraph of the Introduction), which use self-supervised, pre-trained vision encoders, should probably be removed.

In the first paragraph of the Discussion, it seems to me that his paper is implicitly claiming that the scenario examined in this paper is more realistic or challenging than that considered in equivariant networks. I disagree that this is the case, as a network that achieves invariance to 3D rotations would perform perfectly in this scenario.

[Additional empirical results]
Ideally, the paper would also include qualitative neuronal activation results for Cars and Shepard-Metzler objects.

[Hyperparameter settings]
The choice of some hyperparameter settings, such as the 10% threshold for for generalizable vs non-generalizable, and the 95th percentile threshold, are not explained in the paper. Some are not reported (e.g. parameter count/architecture of ViT models).

[Results of multiple runs]
While the multiple runs were made for each experiment, we don't see the results of this (except possibly Figure S.1, which is not very clear: is this a different experiment, with 5 runs for each of (a) and (b), are these 2 of the 5 runs of the same experiment?).

---

> ### Author Response · Authors · 2025-07-14
> **Response to Reviewer: Part 1**
>
> We thank the reviewer for their thorough comments on our manuscript, and appreciate the constructive feedback. We present an improved version of the paper after addressing the requested changes.
>
> 1. **R:** *Clarify the experimental settings, which experimental settings were used to generate the figures/plots, and undefined notations*
>
>     **A:** We have addressed the requested clarifications:
>
>     - **R:** *Clarify the training task, e.g. is it supervised?*
>
>         **A:** We added the following clarifications in Section 2 Methods:
>         - “For each experiment, we train a DNN from scratch on a supervised instance classification task (i.e., N-way classification where each dimension of the output layer corresponds to one object instance)” (p.5, l.114-115)
>
>     - **R:** *At test-time, are the in-distribution images tested exactly those used during training[...]?”*
>
>         **A:** We added the following text to Section 2 Methods:
>         - ​​”The datasets consist of images of object instances, some of which are seen from all orientations during training (fully-seen) and other instances which are only seen from a subset of orientations during training (partially-seen). Each dataset also contains a validation set of fully-seen images and partially-seen instances at orientations seen during training. The distributions of the validation and train overlap, but no specific images overlap. The test set consists of images of the partially-seen instances at OoD orientations, which are not included during training.” (p. 3, l. 70-75)
>
>         We also added a clarification along these lines in Section 3 Results:
>         - “Note that for fully-seen instances (Fig.5b gray dots), all orientations are in-distribution, including the seed, generalizable and non-generalizable (data comes from the validation set).” (p. 12, l. 377-378)
>
>     - **R:** *How are the ranges for sampling chosen?*
>
>         **A:** We have added a paragraph in Section 2 Methods, titled “In-Distribution Orientations for Partially-Seen Instances” with text that reads, in part,
>         - “We chose several ranges of orientation to be in-distribution for the partially-seen instances, which we refer to as seed orientations. We use the term seed because any generalization to OoD orientations must stem from these orientations. In choosing the seed orientations, we chose small ranges of rotation along two of the three axes, and the full range of rotation along the final axis.” (p. 3, l. 91-95)
>
>         In addition, we added Table 1, which lists the various seeds and their rotation ranges.
>
>     - **R:** *In Figure 3, the  labels are not explained, and it's not clear which architecture(s) this plot was generated from. Plots (e.g. Figure 4b, 4c) are not labeled with what dataset and model the plot was generated from.*
>
>         **A:** The list of architectures used in plotting these figures was indeed omitted. To fix this, we have added the following text in section 2 Methods:
>         - “Figures in the main text are shown only for ResNet, except for Figs. 5b,c, which are scatter plots and therefore don’t require averaging over architectures. The analysis was repeated for all architectures and these results can be found in the supplement. This includes accuracy heatmap visualizations (Fig. S3), average OoD accuracy (Fig. S8), and predictive modeling (Fig. S5). The same conclusions drawn in the main paper may be drawn from the results with other architectures as well.” (p.5, l.130-134)
>
>         Further, to reduce confusion about what the dots represent in Figure 4b (now Figure 5b), we modified the caption to read:
>         - “(b)Each experiment is portrayed by four dots, one for each of the invariance scores (Eq. 6), depicted with different colors. Each experiment is a DNN trained on a specific combination seed, object category and proportion of fully-seen, see Sec. 2.” (p.13)
>
>
>     - **R:** *What size of ViT was used?*
>
>         A: We modified Table 2 (p.5) to note that we use ViT-Base, and also included the size of all the other models we experimented with.
>
> 2. **R:** *Reorganize the paper to put Methods before Results*
>
>     **A:** Following the reviewers’ requests to clarify the methodology and experimental setup in the paper, we have reorganized the layout of the paper to have all methodological descriptions precede the results descriptions, thus moving the Methods section before the Results section. Section 2 Methods now consists of a detailed description of the datasets, experimental setup and detailed descriptions of the analytical approaches we employ to generate results. Section 3 Results was slightly modified for better readability.
>
> *Response continues in the next comment*

---

> > ### Author Response · Authors · 2025-07-14
> > **Response to Reviewer: Part 2**
> >
> > 3. **R:** *Address limitations of this work, e.g. limitation to small networks sizes, limitation to only supervised training, limitation to only examining classification. I think this can be addressed by repeating the experiments on larger networks and self-supervised networks as well as another task.*
> >
> >     A: We agree that results for other settings, including self-supervised, contrastive with text, or segmentations would be very interesting, and a good application for the analytical tools we introduce in the work. We therefore highlighted the relevant paragraph in the Discussion and added the following text:
> >     - “This study reveals discernible patterns in the successes and failures of DNNs across diverse orientations which can be effectively characterized and explained through the analysis of neural activity. This underscores the potential for more comprehensive analyses of DNNs that transcend the conventional approach of solely focusing on average accuracy.This study was limited to a supervised classification setting, and to fairly small DNNs. Nonetheless a promising research avenue is to apply the analytical methods introduced in this paper to other newer models.” (p. 14, l.419-421)
> >
> >     In addition, we clarified, as mentioned above in (p.5, l.130-134) that we ran the same experiments and analyses over a range of architectures and model sizes (ResNet18, DenseNet121, ViT-Base) and all yielded the same results.
> >
> > 4. **R:** *Address potential overclaims:*
> >
> >     - **R:** *The implication that this work may have ramifications for Qwen2-VL and Llama 3-V…should probably be removed.*
> >
> >         **A:** The reviewer’s argument made us realize that these sections were in fact over-claims. As they aren’t really needed to advance the paper’s claims, we have removed them.
> >     - **R:** *[I disagree that] the scenario examined in this paper is more realistic or challenging than that considered in equivariant networks.*
> >
> >         **A:** We agree with the reviewer, and have modified the takeaway from these citations. We have moved the first two citations to the introduction:
> >
> >         - “A large number of previous works have begun to understand the generalization capacities in DNNs to OoD orientations. For example, (Lenc & Vedaldi, 2015; Gruver et al., 2023) investigated the emergence of invariance, and specifically rotational invariance to affine rotations. However, it remains unclear whether and how DNNs generalize to OoD orientations, such as in the task we outlined above.” (p. 3, l. 54-57)
> >
> >         In addition, we moved the second pair of citations to the “Limitations and Future Work” section, and added the following text:
> >         - “Other works, including by Cohen and Welling (Group Equivariant CNN’s (Cohen & Welling, 2016) and Steerable CNN’s (Cohen & Welling, 2017)) induce invariance by construction with modifications to the CNN architecture. These works generalize the translation inductive bias inherent in CNN’s to broader mathematical groups, including rotations. However, these approaches focus on architectural improvements that augment the underlying capabilities of these networks, but don’t investigate whether the vanilla architectures exhibit emergent orientation generalization.” (p.14, l.431-436)
> >
> > 5. **R:** The choice of some hyperparameter settings, such as the 10% threshold for for generalizable vs non-generalizable, and the 95th percentile threshold, are not explained in the paper.
> >     **A:** These hyperparameters are part of our model of invariance, and we selected them to remain aligned with our core hypothesis, namely, that neurons act as feature detectors and that invariance supports generalization. We’ve added the following text to Section 2 Methods:
> >     - “This [10%] threshold for generalizable accuracy is intentionally low, as our goal was to capture challenging out-of-distribution cases where the network performs just above chance. This allows us to include orientations that are borderline in terms of generalization. See Fig. S8 for OoD accuracy partitioned between generalizable and non-generalizable — this partition captures model behavior well." (p.8, l. 249-253)
> >
> >     Similarly, with regard to the the 95th percentile threshold for neuron activation, we added to Section 2 Methods:
> >     - “This threshold ensures that we focus on neurons that are clearly active, reinforcing our interpretation of them as meaningful feature detectors, and it avoids relying on marginal or noisy activations that may not be functionally relevant.” (p. 8, l.269-271)
> >
> >     While a full sensitivity analysis could further clarify these choices, we note that our conclusions were robust to slight variations in these thresholds.
> >
> > *Response continues in the next comment*

---

> > > ### Author Response · Authors · 2025-07-14
> > > **Response to Reviewer: Part 3**
> > >
> > > 6. **R:** *While the multiple runs were made for each experiment, we don't see the results of this*
> > >
> > >     **A:** To clarify how results of multiple runs are shown, we added a paragraph in Section 2 Methods:
> > >     - “We re-run each experiment (tuple of dataset and model architecture) five times, each time randomly sampling the specific instances which comprise the fully-seen and partially-seen sets. For all figures, data come from all available repetitions, except where otherwise noted. Error bars denote one standard deviation from the mean.” (p. 5, l. 146-149)

---

### Review · Reviewer_uetx · 2025-06-19

**Summary Of Contributions:**

Empirical analysis of the ability of visual DNNs to generalise from observed object angles to unobserved ones. As a concrete case of extrapolation, or out-of-distribution generalisation, gaining understanding of when such generalisation succeeds or fails is an important question. Reports some generalisation, which can be attributed to generalisation from observed angles to nearby ones, as well as in-plane generalisation. Tries to uncover single-neuron responses which are associated with such generalisation.

**Audience:**

Yes

**Broader Impact Concerns:**

Not provided and not needed.

**Claims And Evidence:**

Yes

**Requested Changes:**

1. Compare results across different architectures, both quantitively (e.g., compare the mean and variance of some statistic) and qualitatively (e.g., in figures where you pool results across architectures, indicate architecture using different markers). Make it clear if you claim that different architectures are similar or not.
 2. Perform baseline analysis on naive models, as detailed above.
 3. Show how the ability to perform out-of-distribution generalisation compares between your task and previous studies, using different tasks (present the data to support the claim that "average classification accuracy across all OoD orientations, our experiments reproduce previous results").

**Strengths And Weaknesses:**

### Strengths
 * A nice experimental approach to attack an important and open question using a limited, well-defined task.
 * Use of three kinds of interesting stimuli, which both differ from one another and were not included in the pre-training of the models, and thus allow for an analysis of fully-observed and partially-observed object angles.
 * The regression model used allows for arguing what kind of out-of-distribution angles get learned, suggesting the main result regarding the contribution of in-plane and nearby angles to generalisation. It would have been even more interesting if you could show more factors which contribute very little to generalisation to unseen angles.

### weaknesses
 * It is not clear if naive models (without training on the datasets included in this study, after the original training on ImageNet, etc) already demonstrate some non-trivial baseline of correct classification for the object at hand. I expect them not to, but this needs to be established as a control. For example, models may exhibit a "prototype effect," where, at a specific angle, classification accuracy is high due to general image recognition training.
 * It is unclear whether the observed generalisation is similar to what was reported in previous studies. It needs to be established that the current approach discusses the same phenomena as out-of-distribution generalisation and that generalisation to unseen orientations does not behave categorically different.
 * The partitioning of the orientations space into generisable and non-generalise orientations is somewhat arbitrary, and you need to demonstrate it does not change dramatically when changing parameters (thresholds, etc.).
 * It is not clear if you claim that different architectures behave similarly in terms of their ability to perform out-of-distribution, and if this behaviour has similar roots (e.g., similar regression coefficients, or similar neural behaviour).
 * The neural analysis contribution is unclear:
   - It seems to be based on very specific assumptions (something like "generalisation is high activation to both seen and unseen angles");
   - It is using ad-hoc defined statistics "invariance score yields higher values when a neuron fires for both sets of orientations" (e.g., if a neuron has constant high activation, will you count it correctly?);
   - It is limited to "neurons in the penultimate layer";
   - It does not lead to interesting conclusions: the idea that "neurons are feature detectors" was assumed by your analysis, as it treats neurons' responses as binary through the use of thresholding.

---

> ### Author Response · Authors · 2025-07-14
> **Response to Reviewer 1**
>
> Thank you for your positive comments regarding our experimental approach.
>
> Following the reviewers’ requests to clarify the methodology and experimental setup in the paper, we have reorganized the layout of the paper to have all methodological descriptions precede the results descriptions, thus moving the Methods section before the Results section. Section 2 Methods now consists of a detailed description of the datasets, experimental setup and detailed descriptions of the analytical approaches we employ to generate results. Section 3 Results was slightly modified for better readability.
>
>
> 1. **R:** *It is not clear if naive models (without training on the datasets included in this study, after the original training on ImageNet, etc) already demonstrate some non-trivial baseline of correct classification for the object at hand… Perform baselines on naive models*
>
>     **A:** This comment is well taken. This point was already addressed in our study but we did not communicate it clearly. For this reason, we reorganized the paper contents (as mentioned above) to highlight the experimental setup. In this new section, we added the following:
>     - “For each experiment, we train a DNN from scratch on a supervised instance classification task (i.e., N-way classification where each dimension of the output layer corresponds to one object instance)” (p.5, l.114-115)
>
>     Note also that we did conduct an ablation where we ran the same experiment with models pre-trained on ImageNet, see Supplementary figures S4, S5, S6 and S8.
>
> 2. **R:** *Show how the ability to perform out-of-distribution generalisation compares between your task and previous studies, using different tasks (present the data to support the claim that "average classification accuracy across all OoD orientations, our experiments reproduce previous results").*
>
>     **A:** The reviewer’s comments on this line, “reproduce previous results,” notified us of a miscommunication on our part. We did not intend to convey that we are directly comparing our task against previous studies, and have removed this line from the paper.
>
> 3. **R:** The partitioning of the orientations space into generisable and non-generalise orientations is somewhat arbitrary, and you need to demonstrate it does not change dramatically when changing parameters
>
>     **A:** These hyperparameters are part of our model of invariance, and we selected them to remain aligned with our core hypothesis, namely, that neurons act as feature detectors and that invariance supports generalization. We’ve added the following text to Section 2 Methods:
>     - “This [10%] threshold for generalizable accuracy is intentionally low, as our goal was to capture challenging out-of-distribution cases where the network performs just above chance. This allows us to include orientations that are borderline in terms of generalization. See Fig. S8 for OoD accuracy partitioned between generalizable and non-generalizable — this partition captures model behavior well. (p. 8, l. 249-253)
>
>     While a full sensitivity analysis could further clarify these choices, we note that our conclusions were robust to slight variations in these thresholds.
>
> 4. **R:** *It is not clear if you claim that different architectures behave similarly in terms of their ability to perform out-of-distribution, and if this behaviour has similar roots… Compare results across different architectures, both quantitively*
>
>     **A:** We’ve clarified with more explicit text in Section 2 Methods, that all analysis and results have been replicated for all architectures:
>     - “Figures in the main text are shown only for ResNet, except for Figs. 5b,c, which are scatter plots and therefore don’t require averaging over architectures. The analysis was repeated for all architectures and these results can be found in the supplement. This includes accuracy heatmap visualizations (Fig. S3), average OoD accuracy (Fig. S8), and predictive modeling (Fig. S5). The same conclusions drawn in the main paper may be drawn from the results with other architectures as well.” (p.5, l.130-134)
>
> *Response continues in the next comment*

---

> > ### Author Response · Authors · 2025-07-14
> > **Response to Reviewer: Part 2**
> >
> > 5. **R:** *The neural analysis contribution is unclear:*
> >
> >     - **R:** *It seems to be based on very specific assumptions (something like "generalisation is high activation to both seen and unseen angles");*
> >
> >         **A:** We appreciate the opportunity to clarify. While our model may appear to rely on specific assumptions, it is in fact grounded in empirical evidence. Our core hypothesis (ie. that neurons function as feature detectors and that invariance in activation across conditions supports generalization) is supported by strong predictive performance. The model is not simply assuming that high activation at both seen and unseen angles leads to generalization; rather, it uses this principle to make accurate predictions about generalization behavior.
> >
> >         Thus, these are not arbitrary assumptions, but interpretable components of a model with demonstrable predictive power. The consistency between this hypothesis and the observed data provides evidence in favor of the underlying mechanisms we propose.
> >
> >     - **R:** *It is using ad-hoc defined statistics "invariance score yields higher values when a neuron fires for both sets of orientations" (e.g., if a neuron has constant high activation, will you count it correctly?);*
> >
> >         **A:** We understand the concern, and we clarify that our invariance score is designed to reflect meaningful selectivity and consistency across conditions, not just constant activation. If a neuron were to exhibit uniformly high activation across all orientations, including both generalizable and non-generalizable ones, it would indeed score highly on the invariance metric. However, such neurons would not correlate strongly with generalization performance because they do not distinguish between conditions.
> >
> >         Crucially, we do not observe such "always-active" neurons in practice. The empirical distributions show that neurons with high invariance scores tend to be selective and active for generalizable orientations, but not for non-generalizable ones. In other words, the invariance we measured emerges only when the neuron is both active and discriminative, and this pattern is predictive of generalization. This suggests that our statistic is not ad hoc, but rather captures a property that aligns with both our hypothesis and observed behavior in the network.
> >
> >     - **R:** *It is limited to "neurons in the penultimate layer"*
> >
> >         A: We added the following text in Section 2 Methods:
> >         - “We focus on neurons in the penultimate layer of the network, which are attuned to the highest level features in the input stimuli, but reflect a consolidated representation of the entire network for inferring the downstream task (instance classification in our simulations).” (p. 7, l. 226-229)
> >     - **R:** *It does not lead to interesting conclusions: the idea that "neurons are feature detectors" was assumed by your analysis, as it treats neurons' responses as binary through the use of thresholding.*
> >
> >         **A:** A model is, by nature, a simplified abstraction of the world designed to provide predictive and explanatory power. In our case, the model aims to capture how invariance in neural activity relates to generalization. Thresholding to quantify invariance is in fact a common technique in the literature; see for example the paper: "Ian Goodfellow, Honglak Lee, Quoc V. Le, Andrew Saxe, Andrew Y. Ng “Measuring Invariances in Deep Networks” NeurIPS 2009."
> >
> >         What sets our work apart is that our model can predict with high accuracy whether a neural network will correctly classify an object in an orientation that is OoD, using only internal neural activations as input. To our knowledge, this level of predictive power is unprecedented. Moreover, our model offers an interpretable and intuitive framework: generalization is supported by invariant and discriminative neural responses.
> >
> >         To further highlight the novelty and utility of our approach, we compared it with standard techniques such as t-SNE. While t-SNE is useful for visualizing neural activity, it does not provide the same predictive insight or quantitative conclusions about generalization that our model offers. See caption of Fig.S9 and in the revised manuscript:
> >         - “Previous neural analysis approaches, like t-SNE, fail to capture the shared representations between fully-seen and partially-seen instances, and therefore cannot be used to advance hypotheses related to dissemination (see Fig.S9).” (p. 12, l. 359-361)
> >
> >         Further, we reformulated the final paragraph of Section 3 Results to highlight the conclusion of our analysis:
> >         - “Taken all together, the correlations between fully-seen invariance and partially-seen invariance, and between partially-seen invariance and generalization behavior, provide substantial evidence that OoD generalization is driven by dissemination of internal network invariances from fully-seen to partially-seen instances through shared features.” (p. 14, l. 397-400)

---

### Review · Reviewer_hW1z · 2025-06-22

**Summary Of Contributions:**

This study empirically investigates the ability of various deep network architectures (CNN, transformers) to generalize invariance to 3D object pose. Networks are trained on an object recognition task (distinguishing planes, cars and Shepard&Metzler shapes). Some objects are seen in all views during training, while others only in some views. Network performance is then assessed at test time on the partially viewed objects. Some transfer of pose invariance is observed, both at the level of single neurons in the penultimate layer of the network and when looking at classification accuracy. This transfer of invariance can both be seen within classes and across classes (e.g., only planes are seen in all views during training, and then the models are tested on Metzler shapes seen in only some views). Interestingly, networks are found to generalize much better in-image-plane rotations than in-depth (3D) rotations.

**Audience:**

Yes

**Broader Impact Concerns:**

N/A.

**Claims And Evidence:**

Yes

**Requested Changes:**

Please clarify the following aspects (or indicate the relevant passage in the manuscript):

1. I am not completely sure what task the networks were trained on. Is it a three-way classification task (plane/cars/SM)?

2. It is unclear and I don't know whether it is said somewhere whether the partially seen views include all alpha, beta and gamma angles together, or whether these angle ranges are explored in separate experiments.

3. In fig 3a, I am not sure to understand why there is an increase in generality for both alpha and gamma. Shouldn't it be only for alpha (rotation in plane)? Also, what do these different conditions represent exactly? Do they represent different angle *training* ranges, or different angle *testing* ranges? And, what is alpha'?

4. In fig 4b, "Each dot represents the results of an experiment". It is unclear to me whether these experiments correspond to networks *trained* on different objects/views or *tested* on different objects/views.

5. Fig 4b and c: labels and ticks are too small. Fig s3: typo in "meaingfully". Fig s6: Labels (a. and b.) missing from the figure.

6. I did not fully understand what the "silhouette" variable is.

**Strengths And Weaknesses:**

I found the study interesting and well executed. The conclusions are also well supported by the results. Some technical aspects of the study could be described more clearly (see requested changes).

Additional remarks:
"A key question arising from our results is to explain why DNNs disseminate orientation-invariance only to in-plane orientations. "
=> I think the convolutional architectures (and patchify process in ViT) may provide an inductive bias helpful for generalizing in-plane rotation but not in-depth rotation. Indeed, in-plane rotations translate (but also rotate) features on the image grid, which convolutional architectures can partly exploit.

Consider updating your citations with more recent work investigating the ability of recent and SoTA architectures (including VLMs) to generalize pose invariance, and comparing them to humans:
Abbas & Deny 2023: https://arxiv.org/abs/2207.08034
Ollikka et al 2025: https://openreview.net/forum?id=yzbAFf8vd5
Michalkiewicz et al 2024: https://arxiv.org/abs/2412.19920
O'Connel et al 2025: https://direct.mit.edu/opmi/article/doi/10.1162/opmi_a_00189/128124/Approximating-Human-Level-3D-Visual-Inferences
Kosoy et al 2025: https://www.arxiv.org/abs/2503.03840

---

> ### Author Response · Authors · 2025-07-14
> **Response to Reviewer**
>
> Thank you for your positive comments regarding your interest in the paper and our execution.
>
> Your citation suggestions were greatly appreciated. We’ve added them in the introduction and the discussion.
>
> Following the reviewers’ requests to clarify the methodology and experimental setup in the paper, we have reorganized the layout of the paper to have all methodological descriptions precede the results descriptions, thus moving the Methods section before the Results section. Section 2 Methods now consists of a detailed description of the datasets, experimental setup and detailed descriptions of the analytical approaches we employ to generate results. Section 3 Results was slightly modified for better readability.
>
> In particular, the reviewer’s clarification questions were very helpful in exposing gaps in our presentation. We’ve modified the relevant sections of the paper to improve readability.
>
> **Answers to requested changes:**
>
> 1. **R:** *I am not completely sure what task the networks were trained on. Is it a three-way classification task (plane/cars/SM)?*
>
>     **A:** We added the following clarifications in Section 2 Methods:
>     - “For each experiment, we train a DNN from scratch on a supervised instance classification task (i.e., N-way classification where each dimension of the output layer corresponds to one object instance)” (p.5, l.114-115)
>     - “We experimented with datasets where both fully-seen and partially-seen instances are from the same category, and also where they come from different categories. For example, where the fully-seen instances were airplanes, and the partially-seen instances were Shepard&Metzler objects.” (p.3, l.86-88)
>
> 2. **R:** *It is unclear and I don't know whether it is said somewhere whether the partially seen views include all alpha, beta and gamma angles together, or whether these angle ranges are explored in separate experiments.*
>
>     **A:** To further clarify the definition of the “seed”, we added the following to Section 2 Methods:
>     - “We chose several ranges of orientation to be in-distribution for the partially-seen instances, which we refer to as seed orientations. We use the term seed because any generalization to OoD orientations must stem from these orientations. In choosing the seed orientations, we chose small ranges of rotation along two of the three axes, and the full range of rotation along the final axis.” (p.3, l.91-95)
>
>     Table 1, which lists the various seeds and their rotation ranges. (p.4)
>
>
> 3. **R:** *In fig 3a, I am not sure to understand why there is an increase in generality for both alpha and gamma. Shouldn't it be only for alpha (rotation in plane)? Also, what do these different conditions represent exactly? Do they represent different angle training ranges, or different angle testing ranges? And, what is alpha'?*
>
>     **A:** The above improved definition of “seed” should hopefully clarify Figure 3a (now Figure 4a). We also included a clarification in the figure’s caption, which reads:
>     - “(a) Network generalization for OoD orientations increases with increasing number of fully seen (blue shading.) This trend holds across in-distribution orientations (seed) and object category conditions – including when fully-seen and partially-seen are the same category, and when they differ (ie. Airplane -> SM).” (p.11)
>
>
> 4. **R:** *In fig 4b, "Each dot represents the results of an experiment". It is unclear to me whether these experiments correspond to networks trained on different objects/views or tested on different objects/views.*
>
>     **A:** To reduce confusion about what the dots represent in Figure 4b (now Figure 5b), we modified the caption to read"
>     - “(b)Each experiment is portrayed by four dots, one for each of the invariance scores (Eq. 6), depicted with different colors. Each experiment is a DNN trained on a specific combination seed, object category and proportion of fully-seen, see Sec. 2.” (p.13)
>
>
> 5. **R:** *Fig 4b and c: labels and ticks are too small. Fig s3: typo in "meaingfully". Fig s6: Labels (a. and b.) missing from the figure.*
>
>     **A:** Thank you for pointing out the typos and readability improvements for the figures. We have made the changes you suggested.
>
>
> 6. **R:** *I did not fully understand what the "silhouette" variable is.*
>
>     **A:** We’ve improved the definition of the “silhouette” component with the following text:
>     - “We refer to the object’s silhouette as a featureless solid shape with its edges matching the outline of the object seen as a shadow from the camera. The third component of the model captures orientations in which the object’s silhouette is similar to the silhouette of the object at the in-distribution views — for example, the similarity in the appearance of an airplane’s silhouette when viewed from above and from below.“ (p.7, l.199-203)

---

> > ### Comment · Reviewer_hW1z · 2025-07-14
> > **Clarity concerns addressed**
> >
> > Thank you, these precision address my clarity concerns except for one: what is alpha' in fig 4a?

---

> > > ### Author Response · Authors · 2025-07-15
> > > **Clarifying alpha'**
> > >
> > > alpha' is one of the seeds. Its parameters are enumerated in Table 1. To explain the choice of this particular seed, we will add the following text to the end of the paragraph (p. 9, l. 304-310):
> > > - See also Fig. S1b, a heatmap where the seed is in the “hole” ($\hat \alpha$ seed). This heatmap yields an inverse of the figure “8” pattern. This heatmap demonstrates a case of "silhouette" generalization, as the airplane is only seen from the front for partially-seen instances, but the model generalizes to views of the airplane from the back.

---

> > > > ### Comment · Reviewer_hW1z · 2025-07-15
> > > >
> > > > OK

---

### Decision · Action_Editor_GY5o · 2025-07-23

**Recommendation:** Accept as is

**Audience:**

Yes

**Audience Explanation:**

The reviewers and I agree that the results will be of interest to various researchers, e.g. those interested in understanding the inductive biases of vision models w.r.t. 3D geometry, or those working on applications in 3D perception.

**Claims And Evidence:**

Yes

**Claims Explanation:**

All reviewers agree that, after revision, the paper meets the standards of claims; I concur. This is a thoughtfully designed study.